# Secondary ossification center induces and protects growth plate structure

**Meng Xie[1], Pavel Gol'din[2], Anna Nele Herdina[1,3], Jordi Estefa[4], Ekaterina V Medvedeva[5], Lei Li[1], Phillip T Newton[1,6], Svetlana Kotova[5,7], Boris Shavkuta[5], Aditya Saxena[8], Lauren T Shumate[9], Brian D Metscher[10], Karl Großschmidt[11], Shigeki Nishimori[9], Anastasia Akovantseva[12], Anna P Usanova[5], Anastasiia D Kurenkova[5], Anoop Kumar[13], Irene Linares Arregui[14], Paul Tafforeau[15], Kaj Fried[16], Mattias Carlström[1], András Simon[13], Christian Gasser[14], Henry M Kronenberg[9], Murat Bastepe[9], Kimberly L Cooper[8], Peter Timashev[5,7,12,17], Sophie Sanchez[4,15,18], Igor Adameyko[1,19], Anders Eriksson[20], Andrei S Chagin[1,5]\***

[1]Department of Physiology and Pharmacology, Karolinska Institutet, Stockholm, Sweden; [2]Department of Evolutionary Morphology, Schmalhausen Institute of Zoology of NAS of Ukraine, Kiev, Ukraine; [3]Division of Anatomy, MIC, Medical University of Vienna, Vienna, Austria; [4]Science for Life Laboratory and Uppsala University, Subdepartment of Evolution and Development, Department of Organismal Biology, Uppsala, Sweden; [5]Institute for Regenerative Medicine, Sechenov University, Moscow, Russian Federation; [6]Department of Women's and Children's Health, Karolinska Institutet and Astrid Lindgren Children's Hospital, Karolinska University Hospital, Solna, Sweden; [7]Semenov Institute of Chemical Physics, Moscow, Russian Federation; [8]Division of Biological Sciences, University of California San Diego, San Diego, United States; [9]Endocrine Unit, Department of Medicine, Massachusetts General Hospital and Harvard Medical School, Boston, United States; [10]Department of Theoretical Biology, University of Vienna, Vienna, Austria; [11]Bone and Biomaterials Research, Center for Anatomy and Cell Biology, Medical University of Vienna, Vienna, Austria; [12]Institute of Photonic Technologies, Research center "Crystallography and Photonics", Moscow, Russian Federation; [13]Department of Cell and Molecular Biology, Karolinska Institutet, Stockholm, Sweden; [14]Department of Solid Mechanics, KTH Royal Institute of Technology, Stockholm, Sweden; [15]European Synchrotron Radiation Facility, Grenoble, France; [16]Department of Neuroscience, Karolinska Institutet, Stockholm, Sweden; [17]Chemistry Department, Lomonosov Moscow State University, Leninskiye Gory 1-3, Moscow, Russian Federation; [18]Sorbonne Université – CR2P – MNHN, CNRS, UPMC, Paris, France; [19]Department of Neuroimmunology, Medical University of Vienna, Vienna, Austria; [20]Department of Mechanics, KTH Royal Institute of Technology, Stockholm, Sweden

**\*For correspondence:**
andrei.chagin@ki.se

**Competing interests:** The authors declare that no competing interests exist.

**Abstract** Growth plate and articular cartilage constitute a single anatomical entity early in development but later separate into two distinct structures by the secondary ossification center (SOC). The reason for such separation remains unknown. We found that evolutionarily SOC appears in animals conquering the land - amniotes. Analysis of the ossification pattern in mammals with specialized extremities (whales, bats, jerboa) revealed that SOC development correlates with the extent of mechanical loads. Mathematical modeling revealed that SOC reduces mechanical stress

within the growth plate. Functional experiments revealed the high vulnerability of hypertrophic chondrocytes to mechanical stress and showed that SOC protects these cells from apoptosis caused by extensive loading. Atomic force microscopy showed that hypertrophic chondrocytes are the least mechanically stiff cells within the growth plate. Altogether, these findings suggest that SOC has evolved to protect the hypertrophic chondrocytes from the high mechanical stress encountered in the terrestrial environment.

## Introduction

The mammalian skeleton articulates via articular cartilage, covering the ends of long bones, and grows in length via epiphyseal cartilage, presented as tiny discs of chondrocytes referred to as growth plates and located near the ends of long bones. The growth plate chondrocytes proliferate, align in the longitudinal direction, and then undergo several-fold enlargement (hypertrophy). Thereafter, the hypertrophic chondrocytes undergo apoptosis or trans-differentiation (*Yang et al., 2014*), leaving their calcified extracellular matrix as a scaffold on which invading blood vessels and osteoblasts form new bone tissue, a process known as endochondral bone formation. The process of endochondral bone formation predominantly depends on chondrocyte hypertrophy, and is very conserved throughout the evolution of vertebrates, being described for fin-rayed fish, lobe-finned fish, stem tetrapods, and modern humans (*Hall, 2015*; *Sanchez et al., 2014*; *Sanchez et al., 2016*).

In humans and most experimental models commonly employed to study bone growth (i.e., mice, rats, and rabbits), the growth plate is separated from the articular cartilage by a bony fragment, called a bony epiphysis, generated developmentally from a secondary ossification center (SOC). In rodents and humans, this skeletal element forms during early postnatal development and splits the initially contiguous cartilaginous element (the cartilaginous epiphysis) into distinct articular and growth plate cartilage (*Hall, 2015*) (see *Figure 1—figure supplement 1A–B* for orientation). Interestingly, this spatial separation of articular and growth plate structures is not always required for articulation or bone growth, since it is absent in several vertebrate taxa, such as stem tetrapods (*Sanchez et al., 2014*; *Sanchez et al., 2016*), Chelonians, Crocodilians, and Urodeles (salamanders) (*Fröbisch and Shubin, 2011*; *Haines, 1938*) (see examples of salamander cartilage in *Figure 1—figure supplement 1C–D*).

These observations raise the intriguing question as to why in many taxa including humans the cartilaginous epiphysis evolved into two functionally and spatially separated structures, one responsible for skeletal articulation and the other one required for skeletal growth. Because both of these structures are involved in numerous pathologies (e.g., growth failure in children and osteoarthritis in seniors), answering this question may be of considerable clinical significance.

Based on the anatomical comparison between the epiphyses of different tetrapods, Haines proposed in 1942 that this separation evolved in evolution to improve the mechanical properties of the cartilaginous joint (*Haines, 1942*). An alternative hypothesis (based on mathematical modeling) suggested that the SOC not precedes but is caused by mechanical stimulations (*Carter and Wong, 1988*). Here we performed evolutionary and comparative analysis, mathematic modeling, physical tests, and biological experiments, which altogether indicate that this separation evolved to protect the growth plate from high mechanical forces associated with weight-bearing in a terrestrial environment.

## Results

### Evolutionary analysis

Confirming previous observations (*Sanchez et al., 2014*; *Sanchez et al., 2016*), 3D microanatomical characterization of the 380-million-year-old lobe-finned fish *Eusthenopteron* revealed longitudinally-oriented trabeculae within the shaft of their humeri (*Figure 1A*), strongly indicating that endochondral ossification facilitates bone elongation in stem tetrapods in a manner similar to that in stem amniotes (290 Mya) (e.g., Seymouria, *Figure 1B*) and present-day mammals (e.g., red squirrel, *Figure 1C*).

Evolutionary analysis revealed that the SOC is not present in anamniotes and first appears in amniotes, animals that relocate their entire lifecycle on land (*Figure 1D*). Among various groups of

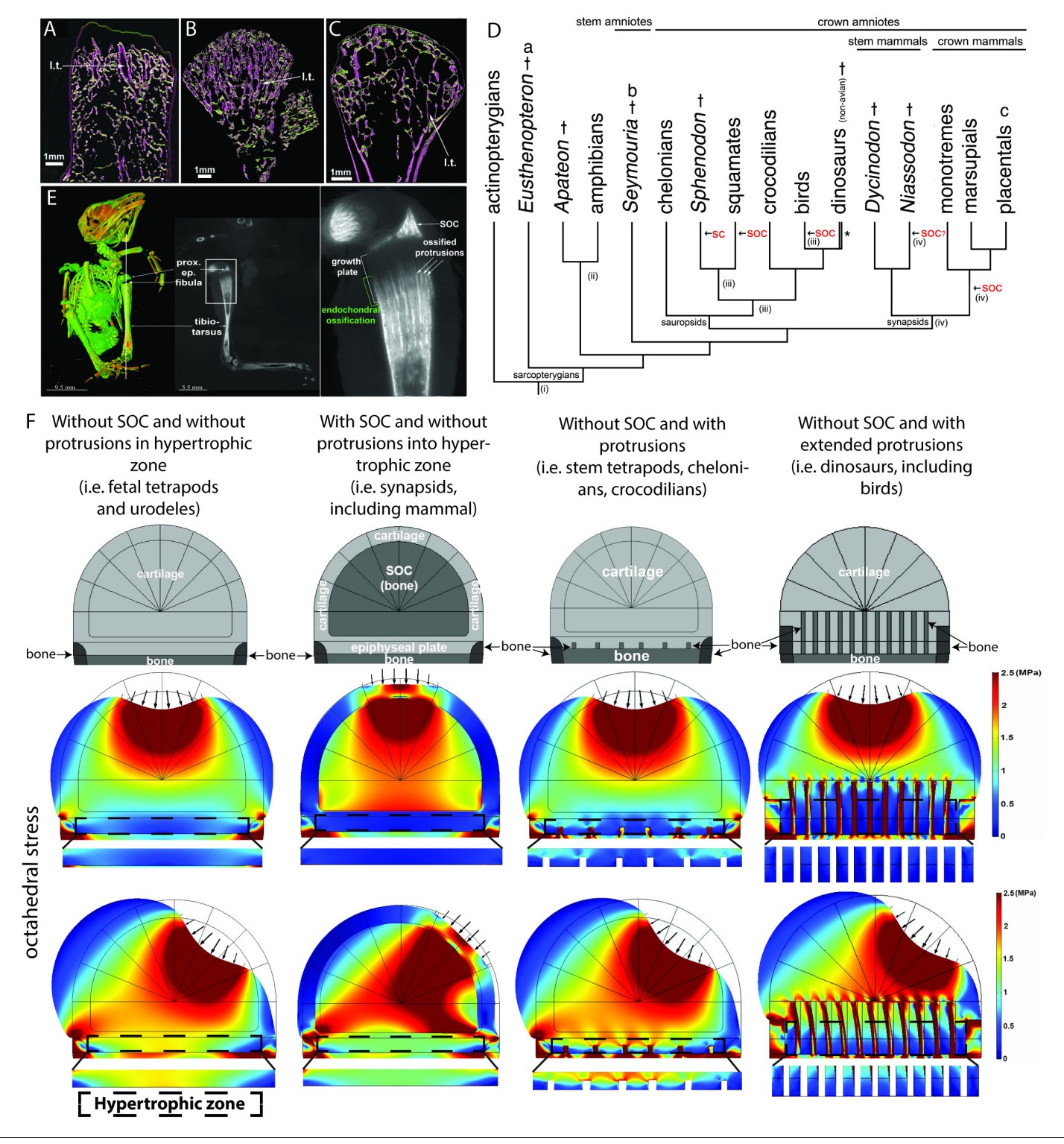

**Figure 1.** Evolutionary appearance of the SOC and its role in distributing octahedral stress. (A–C) The proximal humeral metaphyses of *Eusthenopteron* (A), *Seymouria* (B), and *Sciurus vulgaris* (red squirrel) (C). The longitudinal trabeculae resulting from endochondral ossification at the base of the growth plate in adults are shown in pink, while the transverse trabeculae appear green. The longitudinal trabeculae (l.t.) are designated marrow processes invading the growth plate. (D) Phylogeny tree, illustrating the major changes in the evolution of long-bone epiphyses. The black arrows indicate the presence of a secondary calcified center (SC) or a secondary ossification center (SOC). (i) Estimated appearance of endochondral formation of long bones based on the fossil record to date; (ii) Cartilaginous epiphyses in long-bones; (iii) Hypothetical presence of an ancestral SCs in sauropsids; (iv)

*Figure 1 continued on next page*

*Figure 1 continued*

Hypothetical presence of an ancestral SOCs in synapsids. * indicates paraphyly within the group. (E) A synchrotron scan of a juvenile sparrow demonstrates the presence of SOC. 1, a virtual thin section of the proximal epiphysis. prox. ep., proximal epiphysis. (F) Deformation and distribution of octahedral stress from FEA simulation in different evolutionary taxa. Columns refer to (from left to right) 'without SOC' (representing i.e., fetal tetrapods and urodeles), 'with SOC' (representing i.e., synapsids including mammal), 'stem tetrapods' (representing i.e., juvenile stem tetrapods, chelonians, crocodilians), and archosaurs (representing i.e., birds and non-avian dinosaurs). For comparison of structural functionality, all situations used the same basic geometry and materials. The small arrows indicate the direction of loading. The supra-physiological loading level (3 MPa) was modeled. A hypertrophic zone free of bony elements is presented under each model for direct comparison.

The online version of this article includes the following figure supplement(s) for figure 1:

**Figure supplement 1.** Development of articular and epiphyseal cartilage in rodents and salamanders.

**Figure supplement 2.** Deformation and distribution of comparison stress from FEA simulation in different evolutionary taxa.

synapsids, the SOC is present in all crown mammals, whereas some stem mammals have evolved a SOC [i.e., *Niassodon* (*Castanhinha et al., 2013*; *Figure 1D*)] but others have not (*Chinsamy and Abdala, 2008*; *Ray and Chinsamy, 2004*). In sauropsids, which can be subdivided into two large clades, Lepidosauria (lizards and *Sphenodon*) and Archosauria (crocodilians, chelonians, birds, and non-avian dinosaurs), diversification of bone ossification is greater. Lepidosauria generally develops a SOC (lizards) or calcified secondary center (SC, sphenodontids) (*Figure 1D*). However, Archosauria does not develop such structures [(*Haines, 1938*), *Figure 1D*] with the exception of few bones with SOCs in some birds (*Figure 1E*). Instead, this clade develops bone marrow protrusions with calcified or ossified walls that protrude into the epiphyseal cartilage and sequester the hypertrophic chondrocytes in between them (*Haines, 1938*; *Haines, 1942*; *Barreto et al., 1993*; *Horner et al., 2001*). These ossified protrusions are relatively small in chelonians and crocodilians and quite extended in birds and non-avian dinosaurs (*Haines, 1938*; *Haines, 1942*; *Chinsamy and Abdala, 2008*; *Ray and Chinsamy, 2004* and *Figure 1E*). Although the weight-bearing bones of certain birds exhibit a SOC together with such protrusions (e.g., the proximal epiphysis of the tibiotarsus in sparrow *Figure 1E*), most ossify without forming a SOC (*Horner et al., 2001*).

It has to be emphasized that anamniotes spend their juvenile growth period in an aquatic environment, with amniotes being the first to translocate their entire life cycle on land. Such habitat change may pose additional mechanical demands on the growing skeleton, which must be sufficiently rigid to bear the body weight during a wide range of movements, yet flexible enough to allow bone elongation. Accordingly, we next hypothesized that the development of the SOC or ossified protrusions might be viewed as skeletal adaptations to the weight-bearing demands of the terrestrial environment and associated mechanical stresses posed on the epiphyseal cartilage.

## Mathematical modeling

To explore the potential effects of the SOC and ossified protrusions on stress distribution within the epiphyseal cartilage, we first employed mathematical modeling (i.e., finite element analysis, FEA) based on a modified model developed by Carter and coworkers (*Carter and Wong, 1988*; *Carter et al., 1998*; *Figure 1F*, see the Materials and methods section for details). For description of the local force effects in an equilibrium state, we evaluated the distributions within the model of maximum compressive principal stress, octahedral stress, and hydrostatic stress. These scalar point-wise stress state measures can, in essence, be seen as representing the local stress state by the dominant pressure in any direction (maximum compressive principal stress), the deforming stress tending to deform a spherical region into an ellipsoidal one (octahedral stress), and the uniform shape-preserving pressure tending to change just the size of a sphere (hydrostatic stress). In this evaluation, the shape-changing octahedral shear stress was provisionally assumed to be the most significant one for chondrocytes according to *Carter and Wong, 1988*.

First, the FEA showed that the SOC significantly enhances the stiffness of the entire epiphyseal structure (note the level of deflection in *Figure 1F*), thereby preventing severe distortion and consequent instability during locomotion. This provides experimental support to the idea proposed by *Haines, 1942*. Furthermore, FEA revealed that the presence of either a SOC or ossified protrusions reduces the extent of octahedral shear stress distribution (associated with either a vertical or an angled load, the latter mimicking locomotion) within the region of the epiphyseal plate, particularly where the hypertrophic chondrocytes are located (*Figure 1F*, see other stresses in *Figure 1—figure*

*supplement 2*). Interestingly, despite being similarly protective for epiphyseal chondrocytes, the ossified protrusions do not improve the stiffness of the joint structure. Thus, our mathematical modeling indicates that both of these skeletal adaptations reduce octahedral stress within the epiphyseal cartilage, plausibly representing alternative evolutionary strategies for achieving the same goal.

Thus, in our subsequent analysis, we focused only on the SOC since it is most relevant to human physiology.

## Comparative analysis of animals with specialized extremities

Thus, evolutionary analysis suggests that the appearance of the SOC can be a skeletal adaptation to the terrestrial environment, whereas mathematical modeling suggests that this adaptation may evolve to protect the growth plate structure from mechanical stresses. To further explore the association between mechanical demands and the extent of the SOC development, we selected a few extant unrelated taxa with extremities subjected to different mechanical demands, specifically Chiroptera (bats), Dipodoidea (Jerboa [i.e., bipedal hopping rodents] ), and Cetaceans (whales).

### Chiroptera

The newborn pups of Chiroptera must cling to their mother or to the roost from birth, utilizing for this purpose only their feet, thumbs of the wing, and milk teeth, whereas other wing bones are not used until 2–3 weeks of age, when the pups learn to fly (*Hughes et al., 1995*; *Powers et al., 1991*; *Elangovan et al., 2007*; *Wang et al., 2014*). We observed that in newborn bats the SOC development in the legs and thumbs of wings is more advanced than in the rest of the wing bones

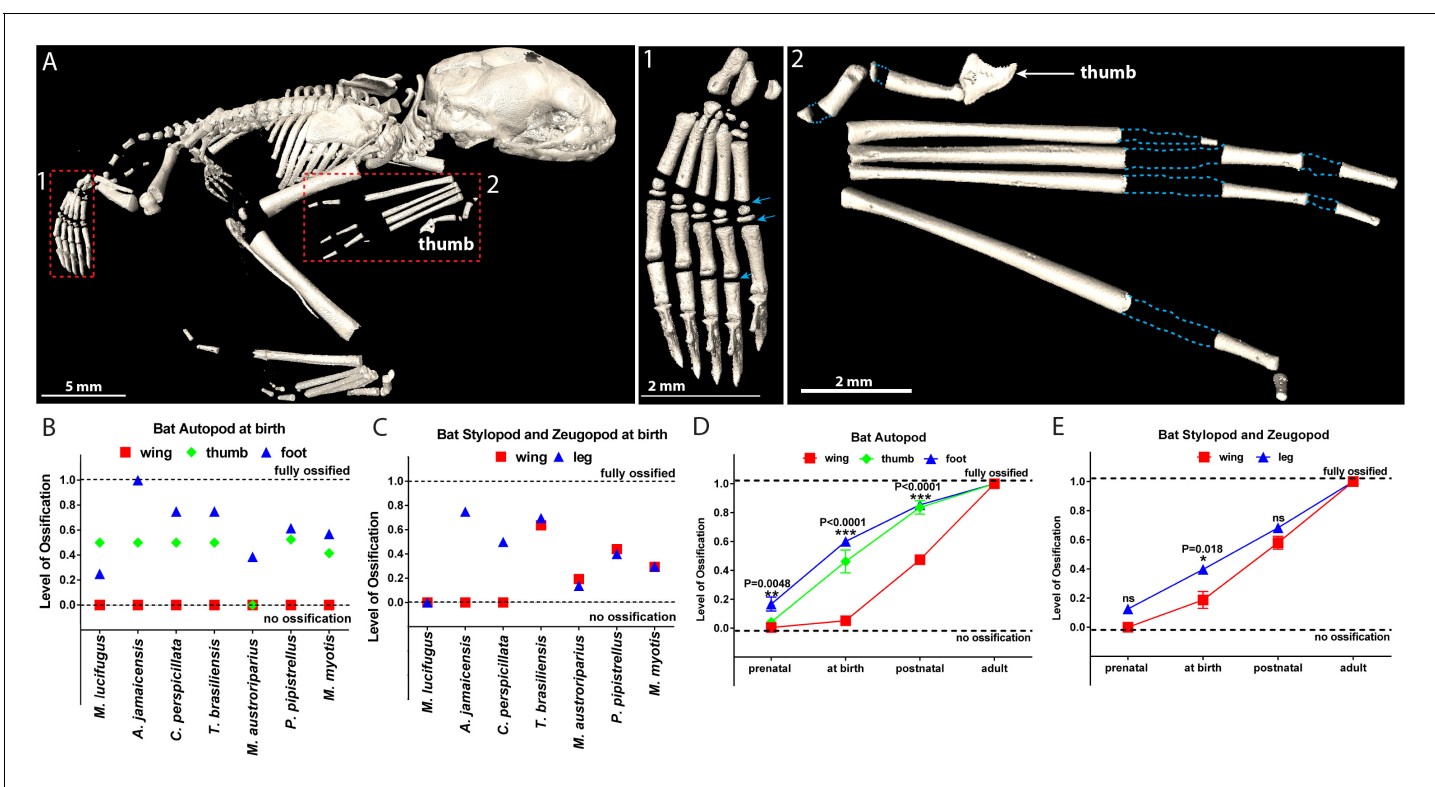

**Figure 2.** Comparative analysis in bats. (**A**) microCT image of a 1–3 days-old bat *Pipistrellus pipistrellus*. (A1) foot, with arrows pointing to growth plates, (A2) wing, with dashed blue lines outlining the areas of non-ossified cartilage. (**B–C**) Ossification of the autopod (**B**) and of the stylopod and zeugopod (**C**) in seven bat species at birth. (**D–E**) A generalized pattern of ossification of the autopod (**D**) and stylopod and zeugopod (**E**) of six bat species combined (*Myotis lucifugus, Artibeus jamaicensis, Carollia perspicillata, Tadarida brasiliensis, Myotis austroriparius*, and *Myotis myotis*).

The online version of this article includes the following video for figure 2:

**Figure 2—video 1.** 3D model based on microCT surface images of a 1–3 day-old *Pipistrellus pipistrellus s.l.* (common pipistrelle bat).
https://elifesciences.org/articles/55212#fig2video1

(*Figure 2A* and *Figure 2—video 1*). This heterochrony, that is, a change in timing or rate of ontogenetic development, was particularly pronounced in autopods with heavily ossified feet and thumb, but no SOC in other bones of the wing (*Figure 2B–C*). It was apparent during ontogenesis and consistent within the Chiroptera order (summary of six species with available ontogeny data is presented in *Figure 2D–E* and primary data for all 13 species analyzed presented in *Supplementary file 1*). It has to be admitted that despite a very clear pattern of heterochrony, there is a variation of ossification among Chiroptera species, which likely reflects their varying foraging and roosting ecology, behavior of juvenile bats, litter size, and lifespan (*Adams and Thibault, 2000*; *Hermanson and Wilkins, 2008*; *Koyabu and Son, 2014*; *Kunz and Hood, 2000*; *Racey and Entwistle, 2000*).

Interestingly, SOCs in the feet and the thumbs of some bat species are well-developed even before birth (*Supplementary file 1*) and the pups of certain species (*Mops condylurus, Artibeus jamaicensis, Megaloglossus woermani,* and *Rousettus celebensis*) are born with fully ossified feet of adult size (*Adams and Thibault, 2000*; *Koyabu and Son, 2014*) (see also *Figure 2B* for *A. jamaicensis*). The appearance of SOC in embryonic feet and thumb suggests that a genetic program governing SOC development precedes the mechanical influence.

## Jerboa

Next, we analyzed SOC development in the bipedal three-toed jerboa (*Jaculus jaculus*) in the family of Dipodoidea rodents. These animals jump on their hindlimbs when adult but during the first 2–3 weeks of life they employ only forelimbs thereafter becoming quadrupedal and eventually bipedal (*Eilam and Shefer, 1997*). The video-recording of gait pattern acquisition for two species, *Jacuslus jaculus* and *Jaculus orientalis,* confirmed that jerboa employs only front limbs for crawling during the first postnatal weeks (*Figure 3—videos 1* and *2*). Analysis of the ossification pattern for *Jacuslus jaculus* revealed that SOC formation in the front limbs significantly preceded the one in the hind limbs (*Figure 3A–R*). It is interesting to point out that the development of the SOC also coincides with the changes in gait pattern from crawling to walking in mice and rats (*Figure 1—figure supplement 1A–B*). Whether the same situation applies to human remains to be investigated.

Thus, in both Chiroptera and jerboa, advanced ossification is observed in limbs subjected to mechanical duties early in life, which is further strengthening an association between mechanical demands and SOC development.

## Cetaceans

If demand for SOC is mechanically related, we further reasoned that the absence of terrestrial-associated loading conditions would release the evolutionary pressure on SOC and that it can be lost or reduced with time. To test this assumption, we analyzed the cetaceans (whales) forelimbs (*Figure 4A*) whose mechanical demands highly differ in loadings and angles from their terrestrial ancestors (*Cooper et al., 2008*). In early cetaceans from the Eocene epoch (56–33.9 Mya), the epiphyseal plates of the forelimbs are structurally similar to those of terrestrial mammals, as evident in semi-aquatic *Maiacetus inuus* (*Gingerich et al., 2009*) and fully aquatic *Dorudon atrox* (*Form, 2004*), where SOCs are present at the joints connecting carpal bones, metacarpal bones, and phalanges (*Figure 4B* 1-2). However, in baleen whales (both extant and some extinct from Miocene epoch, 23–5.3 Mya), there is a clear reduction of SOCs such as in the Balaenopteridae family where SOC is reduced by up to 80% in metacarpals and phalanges (*Figure 4B* 3) and the Balaenidae family where SOC is reduced in distal radius and ulna and essentially absent in metacarpals phalanges (*Figure 4B* 4).

Toothed whales from the Miocene to the present demonstrate more diverse patterns of secondary ossification. SOCs are not reduced in either Platanistidae or certain Delphinoidea (*Figure 4B* 5), whereas Iniidae (Amazon River dolphins) do not have distinct epiphyseal plates, but rather multiple small SOCs covering 10–50% of the corresponding area (*Figure 4B* 6). In Ziphiidae and the narwhal *Monodon monoceros* (Delphinapteridae), SOCs are reduced in size by 5–10 times (*Figure 4B* 7). X-ray examination of another member of Delphinapteridae, the beluga whale *Delphinapterus leucas*, revealed only remnants of SOC in metacarpals and phalanges (*Figure 4D*). Finally, the most advanced epiphyseal transformation is seen in the killer whale *Orcinus orca* (Delphinidae) (*Figure 4B* 8; *Cooper, 2009*; *Mellor et al., 2009*) and the sperm whale *Physeter catodon* (*Figure 4B* 9 and 10), in which the SOCs of the distal radius, ulna, and metacarpals are indistinct or even absent and a

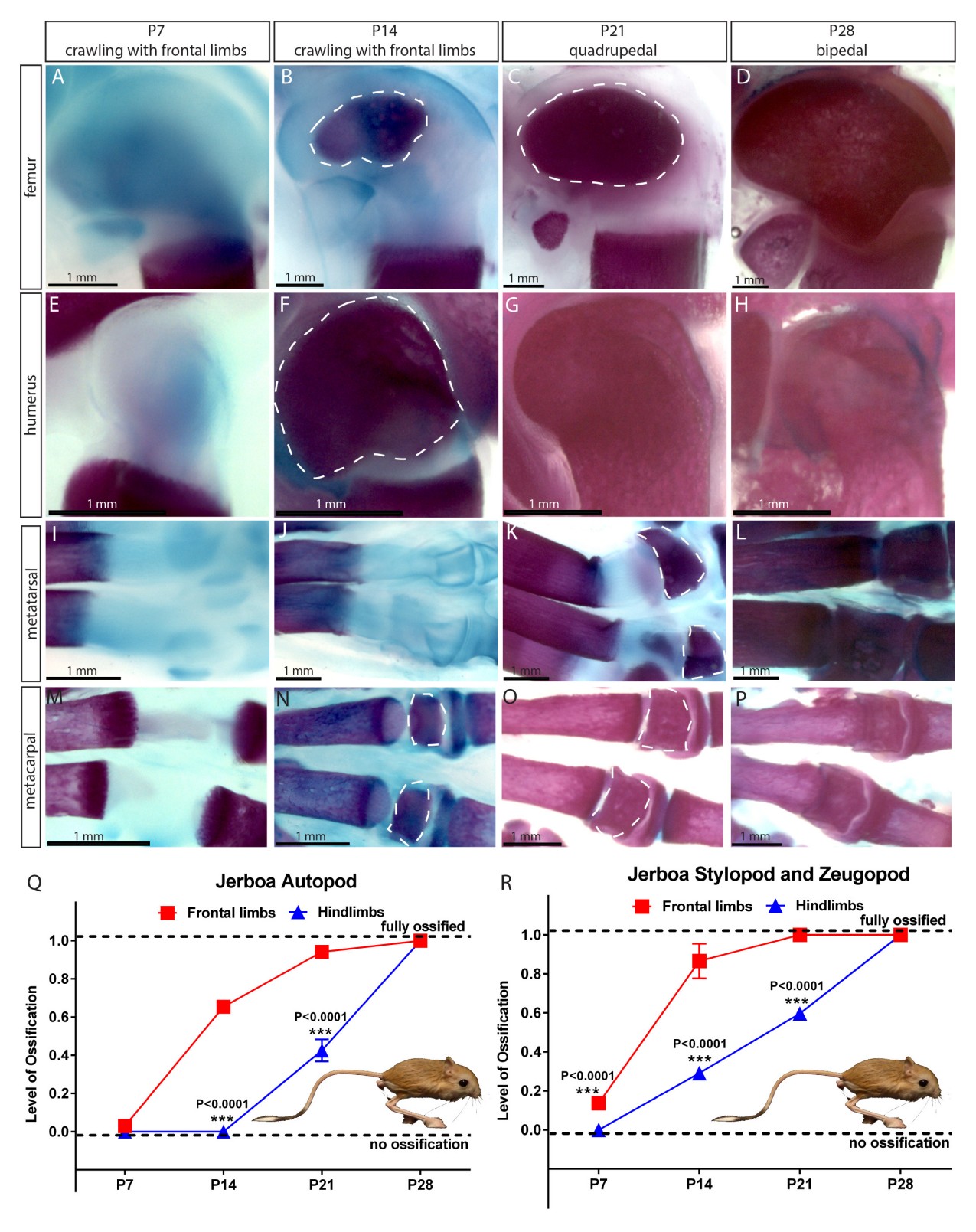

**Figure 3.** Comparative analysis in jerboa. (A–P) Alcian blue and alizarin red staining of the femur (A–D), humerus (E–H), metatarsal (I–L), and metacarpal (M–P) of *Jaculus jaculus* (three-toed jerboa) at various postnatal (P) time points. The dashed white lines outline the SOC. The gait pattern was derived from our weekly observations (see *Figure 3—videos 1* and *2*). (Q–R) Ossification of the autopod (Q) and stylopod and zeugopod (R) of *Jaculus jaculus*

*Figure 3 continued on next page*

*Figure 3 continued*

(three-toed jerboa) at various postnatal stages. Data are means ± SD, representing inter-individual variation, one-way ANOVA. In (**Q–R**), n = 2 jerboa analyzed. ns, not significant.

The online version of this article includes the following video(s) for figure 3:

**Figure 3—video 1.** Development of a gait pattern of jerboa *Jaculus jaculus*.
https://elifesciences.org/articles/55212#fig3video1
**Figure 3—video 2.** Development of a gait pattern of jerboa *Jaculus orientalis*.
https://elifesciences.org/articles/55212#fig3video2

fibrocartilaginous structure is formed in the carpal area (*Figure 4B* 8–10). CT scan confirmed a complete lack of the SOC in *Orcinus orca* (*Figure 4E* and *Figure 4—video 1*), which was further confirmed by radiograph (*Figure 4C*). Taken together, these observations suggest that the return of cetaceans to the aquatic environment is associated with a gradual reduction in the size of SOC in a few phylogenetic lineages, and even their complete loss in some species.

The observed striking consistency in unrelated taxa indicates a causative relationship between the presence of the SOC and mechanical demands and considering mathematical predictions and evolutionary analysis, further suggests that SOC protects the growth plate from mechanical demands.

## Functional experiments with two model systems: the SOC protects epiphyseal chondrocytes

To explore the function of the SOC experimentally, we first employed two ex vivo models: (i) application of different loads to bones of similar size but with (mouse) or without (rat) a SOC and (ii) application of identical pressure (load per area) to rat bones at different stages of development (before, during, and after the formation of the SOC).

We first compared similar rodent tibiae, one with and one without a SOC. Analysis of development pattern of mouse and rat bones revealed that the tibiae from 10-day-old rats and 30-day-old mice are comparable in size and shape (*Figure 5—figure supplement 1A*), as well as in the mechanical properties of the cartilage (*Figure 5—figure supplement 1B*). However, the rat tibia has not yet developed a SOC (referred to here as the SOC- model), whereas the murine tibia does have a well-developed SOC at this age (the SOC+ model) (*Figure 1—figure supplement 1A–B*). We applied vertical or angled loads, which represent the loading conditions at rest or during locomotion, respectively, with different forces to the SOC- and SOC+ models and first analyzed their physical properties. Consistent with the mathematical prediction, the SOC enhanced the stiffness and elasticity of these bones in connection with both vertical and angled loads (*Figure 5—figure supplement 1C–J*). Although confounding variables cannot be definitively excluded, the agreement between the mathematical prediction and the physical tests further support our model that SOC provides additional stiffness, while simultaneously reducing octahedral shear stress within the epiphyseal growth plate.

To address the protective effect of SOC on epiphyseal chondrocytes, we placed the bones exposed previously to different loads into in vitro culture conditions for 48 hr to allow load-induced cellular responses to manifest, using previously optimized culture conditions (*Chagin et al., 2010*). We then assessed the survival status of the epiphyseal chondrocytes using exposure to propidium iodide (PI) during the end of the culture period (i.e., a part of live/death assay to visualize dead cells). These experiments revealed that chondrocytes appeared to be highly sensitive to load, with 40% dying upon application of a 1N-load in the SOC- model; whereas the presence of SOC allowed these cells to withstand a load one order of magnitude higher (*Figure 5A–B* and *Figure 5—figure supplement 2*). The unfavorable effect of loading on epiphyseal chondrocytes in both the SOC- and SOC+ models was confirmed by the decreased expression of several typical chondrocyte markers, such as Indian hedgehog (Ihh), parathyroid hormone-related peptide (PTHrP) (*Figure 5—figure supplement 1K–L*), and collagen type X (Col X) (*Figure 5—figure supplement 1N–O*).

To distinguish between mechanical damage and stress-induced activation of the self-elimination apoptotic program, we performed the TUNEL assay on the loaded bones to detect fragmented DNA, a hallmark of apoptotic nuclei. Epiphyseal chondrocytes in the SOC+ model become apoptotic in response to much higher loads as compared to the SOC- model (*Figure 5C–E*, see also *Figure 5—figure supplement 1M* for baseline control). In addition, much higher loads were required

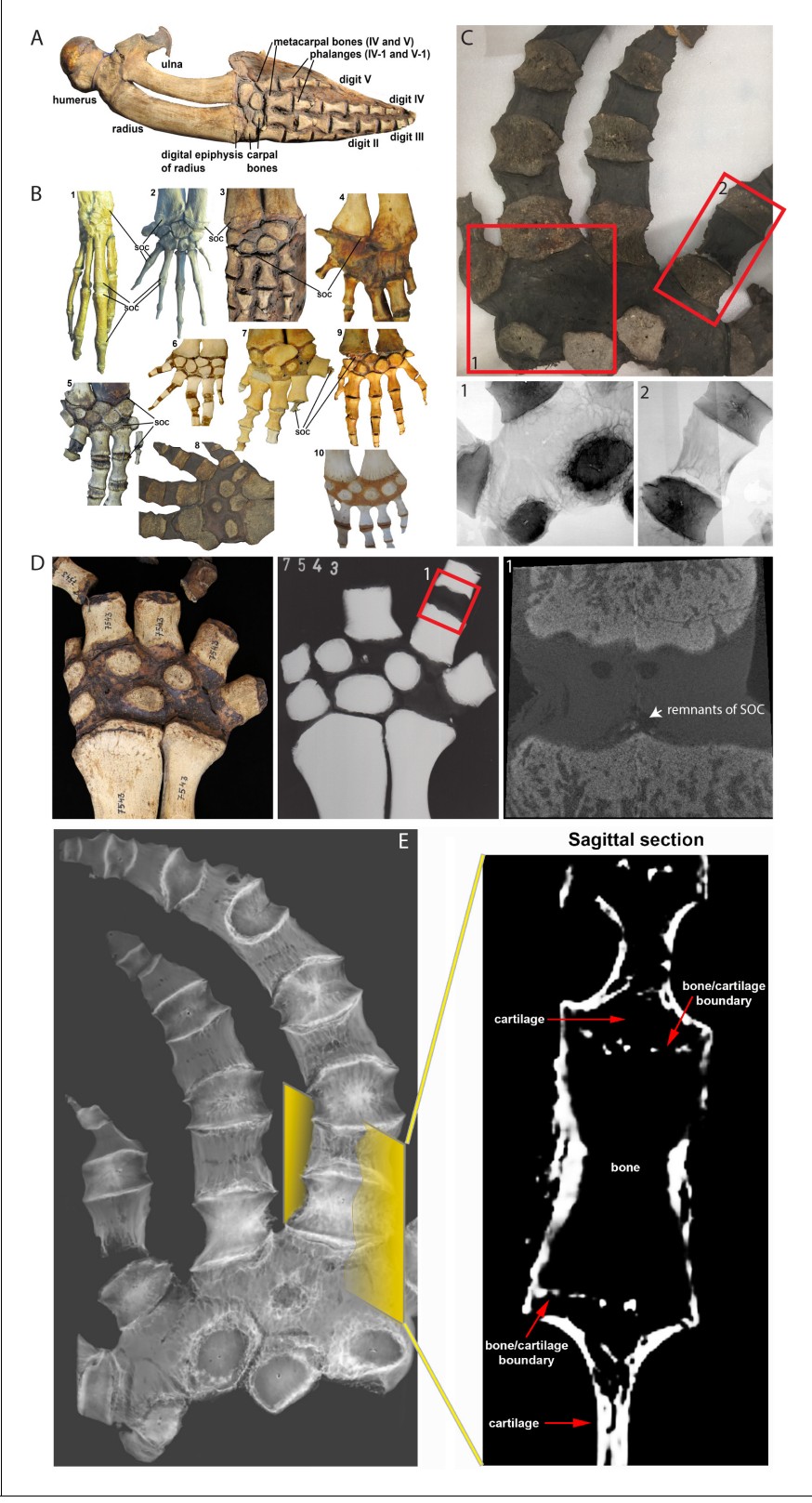

**Figure 4.** Comparative analysis in whales. (**A**) Forelimb bones of the minke whale *Balaenoptera acutorostrata*. (**B**) Comparative anatomy of bony and cartilaginous epiphyses in the distal portion of the forelimbs of various cetacean whales: B1-B2 – early cetaceans; B3-B4 – baleen whales; B5-B10 – toothed whales. (**B1**) *Maiacetus inuus*, semi-aquatic from Eocene era (56–33.9 Mya), (**B2**) *Dorudon atrox*, fully-aquatic from Eocene era (**B3**) minke whale *Balaenoptera acutorostrata* from Miocene era (23–5.3 Mya) (**B4**) North Atlantic right whale *Eubalaena glacialis*, (**B5**) pilot whale *Globicephala melas* (**B6**) *Figure 4 continued on next page*

*Figure 4 continued*

boto river dolphin *Inia geoffrensis*, (B7) narwhal *Monodon monoceros*, (B8) killer whale *Orcinus orca*, (B9) juvenile and (B10) subadult sperm whale *Physeter catodon*. (C) Sample image of an adult killer whale *Orcinus orca* (same species as B8) with X-ray images of the highlighted areas (C1 and C2) from the corresponding image on their left. (D) Sample (left) and X-ray (right) images of an adult beluga whale *Delphinapterus leucas*. (D1) Micro-CT reconstruction of the highlighted phalange joint. (E) CT image of an adult killer whale *Orcinus orca* and a virtual sagittal section of the metacarpal II demonstrating the absence of the SOC.

The online version of this article includes the following video for figure 4:

**Figure 4—video 1.** CT scan of the manus of a killer whale *Orcinus orca.*

https://elifesciences.org/articles/55212#fig4video1

for the SOC+ model to activate caspase-3, a protease that executes apoptosis (*Figure 5F*). To further explore the underlying mechanism, we checked the activity of the signaling pathway involving the Yes-associated protein 1/Tafazzin (YAP/TAZ), which is well-known to be involved in mechano-sensing (*Dupont et al., 2011*) and can promote apoptosis via the tumor suppressor protein, p73 (*Strano et al., 2001*). Both the levels and nuclear translocation of the YAP (*Figure 5—figure supplement 3A* and *Figure 9—figure supplement 1K–L*) and p73 proteins (*Figure 5—figure supplement 3B*) were enhanced by loading, with their overlapping distribution pattern further confirming the activation of this signaling pathway (*Figure 5—figure supplement 3C*). Thus, caspase-dependent apoptosis, likely via the YAP-p73 pathway, appears to be triggered in epiphyseal chondrocytes by mechanical loads with a SOC providing substantial protection.

We also noticed that cell death was most extensive in the hypertrophic zone (*Figure 6A–D*), where the levels of active caspase-3, YAP, and p73 were also increased to a greater extent in the hypertrophic zone than in the resting/proliferative zone (*Figure 6E–H* and *Figure 6—figure supplement 1A–D*). In SOC- bones hypertrophic chondrocytes died in response to a load as low as 0.2N (*Figure 6I*), whereas in the SOC+ bones this phenomenon occurred only with loads of 3–5N (*Figure 6A–D*). Within this loading range, no effect on chondrocyte proliferation was observed (*Figure 6—figure supplement 1E–F*), suggesting that loading affects particularly the hypertrophic chondrocytes.

Despite the agreement with the mathematical predictions, confounding variables cannot be definitively excluded in this SOC ± model. To further strengthen this line of experimental evidence, we compared the effect of the same pressure (load per cross-sectional area) on rat tibiae before, during, and after the formation of SOC. The tibia was isolated from rats of the same litter at 5, 8, 12, 15, and 22 days of age and then measured in order to apply a load (0.2–0.65N) that generated a pressure of 0.023 MPa per cross-sectional area. We found that a pressure identical to that which led to the death of chondrocytes before the formation of the SOC did not affect the survival of these cells after this formation (*Figure 7A–D*). In addition, hypertrophic chondrocytes were proved to be particularly vulnerable to this effect of pressure (*Figure 6E*). The percentage of the hypertrophic chondrocytes that died and the size of the SOC were negatively correlated (*Figure 7F–G*).

Thus, both these ex vivo models suggest that the SOC protects the growth plate chondrocytes from mechanical stress, the effect particularly pronounced in the hypertrophic cell population, which renders these cells able to withstand approximately twenty-five-fold greater load when a SOC is present (comparing 0.2 N to 5N). In combination with mathematical modeling (compare the two left columns in *Figure 1F*), these findings indicate that the SOC protects chondrocytes and, especially, hypertrophic chondrocytes from the deleterious effect of extensive mechanical stress.

## Functional experiments with physiological systems: the SOC protects epiphyseal chondrocytes

This highly effective protection by SOC observed in the ex vivo models prompted us to modulate the size and maturity of the SOC in a series of in vivo experiments to compare the survival status of the epiphyseal chondrocytes. We recently reported that an inhibitor of vascular endothelial growth factor (VEGF) receptor, axitinib (*Gerber et al., 1999*), can delay development of the SOC within a narrow developmental window, that is when axitinib is administrated at P18, but not P21 as analyzed in transgenic mice (*Newton et al., 2019*). Here similar inhibition on the size of SOC was observed in wild-type mice (*Figure 8A–B*). The mice in which the development of the SOC was delayed showed

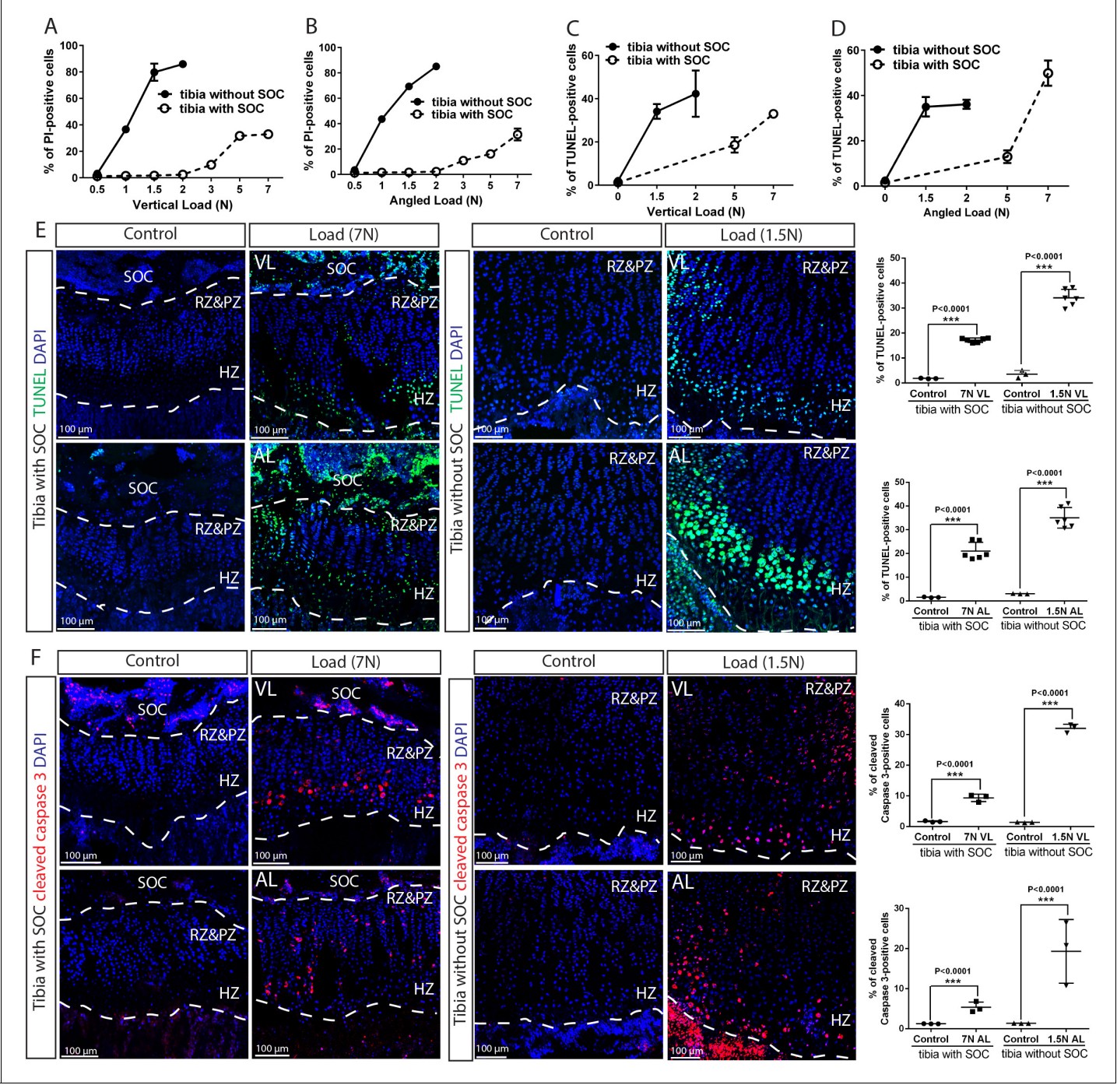

**Figure 5.** The SOC protects epiphyseal chondrocytes from apoptosis induced by mechanical stress. (A–D) Quantification of propidium iodide- (PI) (A–B) and TUNEL- (C–D) positive cells in the growth plates of the tibia with and without SOCs loaded vertically (A, C) or at an angle (B, D). (E–F) Representative images and quantification of TUNEL staining (green) (E) and cleaved caspase-3 staining (red) (F) in the growth plates of loaded tibias with and without SOCs. DAPI was used for counterstaining (blue). Data are means ± SD, two-tailed Student's t-test. In (A–F), n = 3. The control and loaded tibia were from the same animal. VL, vertical load, AL, angled load. RZ and PZ, resting zone and proliferative zone combined, HZ, hypertrophic zone. 'tibia with SOC' and 'tibia without SOC' in the figure refer to tibias from 30-day-old mice and 10-day-old rats, respectively (see **Figure 5—figure supplement 1A–B**).

The online version of this article includes the following figure supplement(s) for figure 5:

**Figure supplement 1.** Establishing and characterizing SOC± model in relation to stiffness and cell death.

**Figure supplement 2.** Death of the growth plate chondrocytes upon various loading in the presence and absence of the SOC.

**Figure supplement 3.** YAP/p73 activation the growth plate chondrocytes upon loading.

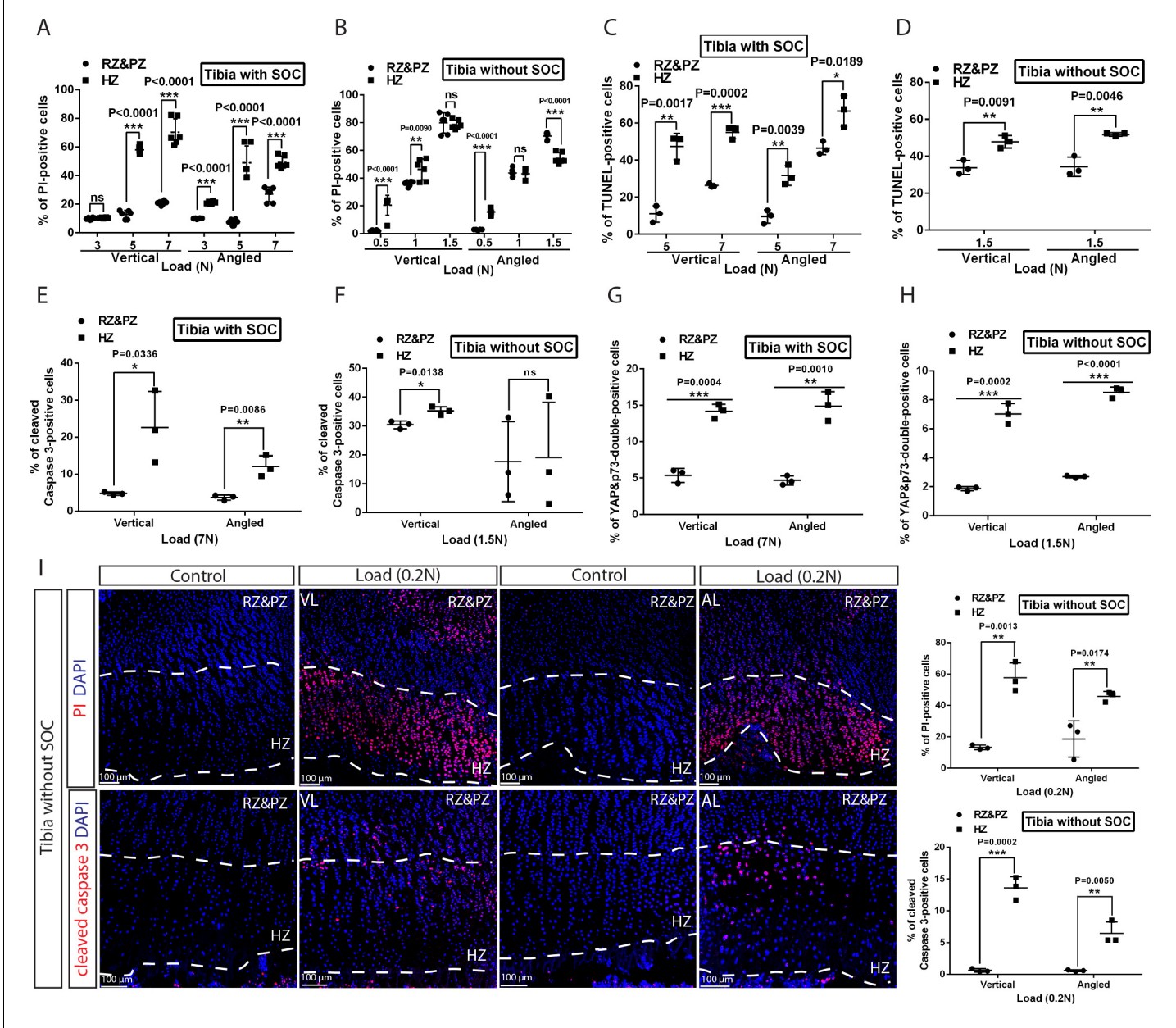

**Figure 6.** Hypertrophic cells are the most sensitive cell type in response to mechanical stress. (A–D) Comparison of the extent of cell death between resting+proliferative zones (RZ and PZ) and hypertrophic zone (HZ) of the vertically and angularly loaded growth plates with (A, C) and without (B, D) SOCs as assessed by PI staining (A–B) and TUNEL (C–D). (E to H) Distribution of immunohistological staining along the longitudinal axis for cleaved caspase-3 staining (E–F) and YAP and p73 double staining (G–H) of the vertically and angularly loaded tibia with (E, G) and without (F, H) SOCs. (I) Representative images and quantification of propidium iodide (PI) staining (upper panel) and cleaved caspase-3 staining (lower panel) in the growth plates of vertically or angularly 0.2N loaded tibias without SOCs. The control and loaded tibia were from the same animal. DAPI was used for counterstaining (blue) in (I). Data are means ± SD, two-tailed Student's t-test. In (A–I), n = 3. ns, not significant. 'tibia with SOC' and 'tibia without SOC' refer to tibias from 30-day-old mice and 10-day-old rats, respectively (see *Figure 5—figure supplement 1A–B*).

The online version of this article includes the following figure supplement(s) for figure 6:

**Figure supplement 1.** Distribution of YAP/p73 activation along longitudinal bone axis and the effect of loading on chondrocyte proliferation.

significantly higher levels of apoptosis in the hypertrophic zone than corresponding controls (*Figure 8C–D*). Axitinib treatment until the same final time point (P27) or for the same duration (P21-P30) was employed as controls in these experiments (*Figure 8A,C*). No significant effect of axitinib on mechanical properties of epiphyseal cartilage was observed as assessed by nanoindentation

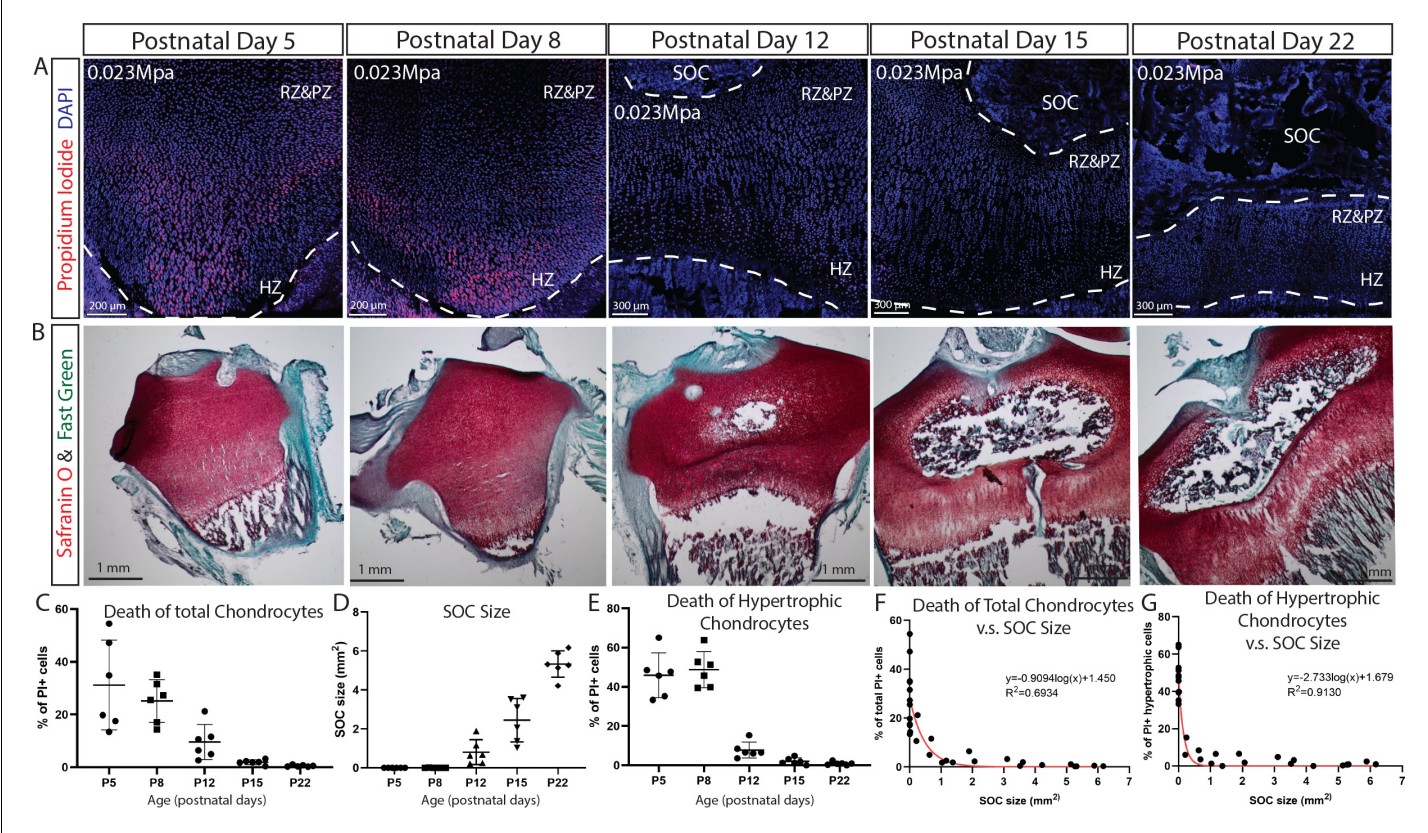

**Figure 7.** SOC size is negatively correlated with load-induced chondrocyte death in rat tibia growth plate. (**A**) Representative images of the distribution of propidium iodide (PI)-positive cells in the growth plates of rat tibia with the same vertical pressure of 0.023 Mpa (equivalent to the load of 0.2N for postnatal day 5 rat tibia). (**B**) Representative images of safranin O and fast green staining of the corresponding rat tibia to illustrate the development of SOC. (**C–E**) Quantification of PI-positive total growth plate chondrocytes (**C**), SOC size (**D**) and PI-positive hypertrophic chondrocytes (**E**) in the loaded rat tibia at different ages. (**F–G**) Correlation analysis of PI-positive total growth plate chondrocytes (**F**) or PI-positive hypertrophic chondrocytes (**G**) with SOC size. Semilog line of nonlinear regression curve fit was used (red line). During the end of the culture period, bones were incubated with PI, which penetrates only into dead cells (red). DAPI (blue) counterstaining of all nucleus was done on the fixed tissue sections. RZ and PZ, resting zone and proliferative zones, HZ, hypertrophic zone. Data are means ± SD, n = 6. Two independent experiments were performed with three animals per time point per experiment. Each data point in (**F–G**) represents one animal.

(8.1 ± 1.7 kPa (mean ± SE, n = 6, control animals) and 10.5 ± 1.9 kPa (n = 7, axitinib-treated animals), p=0.37, unpaired t- test; mice were treated with vehicle or axitinib from P21 till P30 and live tissue sections were analyzed).

In a second model, we employed mice with limb-specific activation of the stimulatory G-protein α-subunit (*Gnas*), which exhibit delayed SOC formation (*Prx-Cre:Gnas^R201H^*) (*Karaca et al., 2018*). To block immediate elimination of dead hypertrophic chondrocytes by ingrowing blood vessels, we inhibited angiogenesis in these mice and corresponding controls by injecting axitinib at a time-point when it does not interfere with the formation of the SOC (P21-P30, *Figure 8A*) but delays elimination of hypertrophic chondrocytes (*Newton et al., 2019*). Analysis of these mice confirmed reduced SOC size in mutant mice and revealed an elevated number of dead hypertrophic chondrocytes in these mice (*Figure 8E–F*). It must be emphasized that in both these pharmacological and genetic models, mice actively run and thereby expose their growth plates to the weight-associated loads, but have decreased body mass (for axitinib-treated mice, 56.6 ± 11.2% of the DMSO-treated control and for the *Prx-Cre:Gnas^R201H^* mice, 80 ± 6.9% of the corresponding controls). Thus, the increase in cell death observed may be an underestimation of what might occur in bones with an underdeveloped SOC under normal weight-bearing conditions.

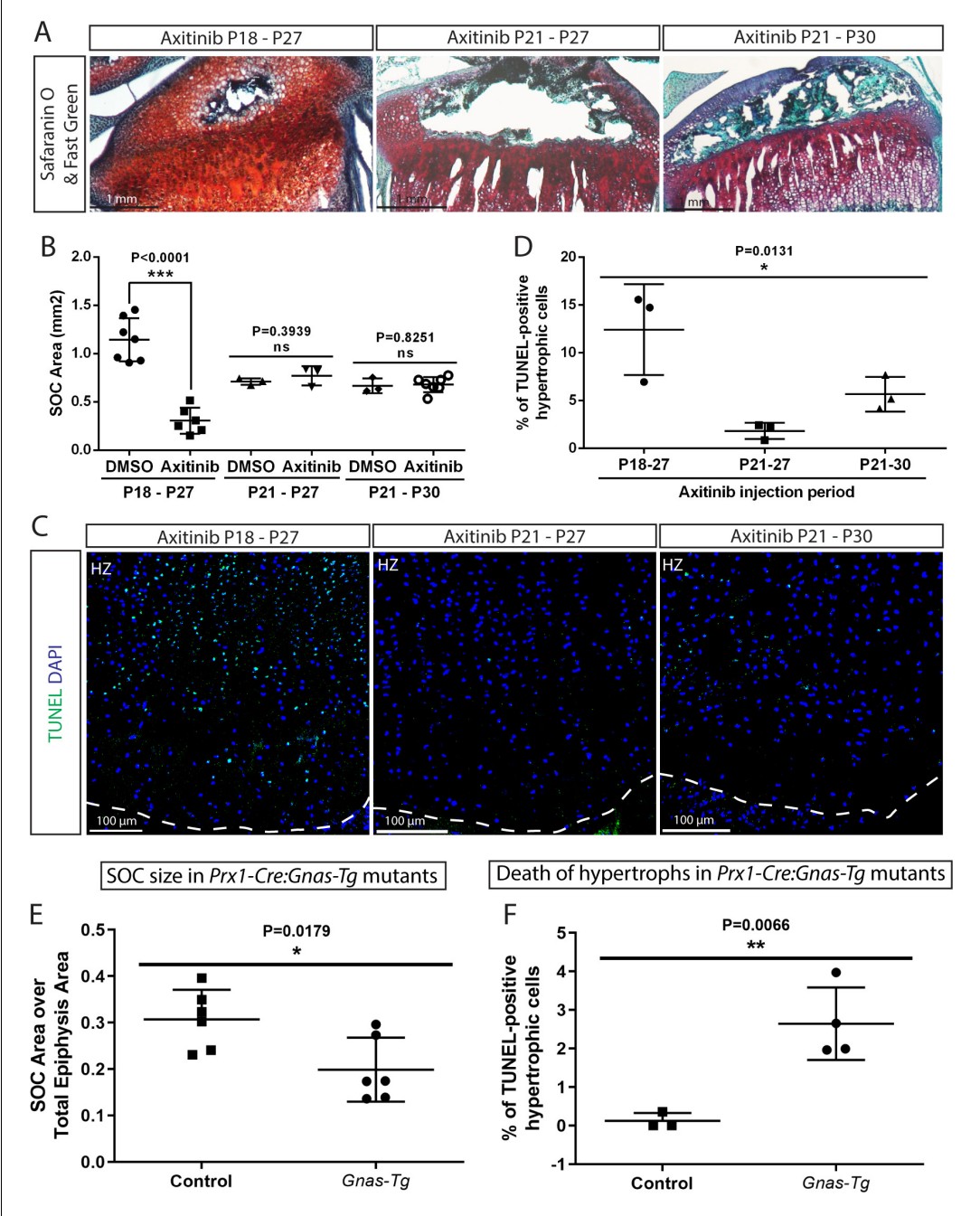

**Figure 8.** Inhibition of the SOC size in vivo is associated with increased epiphyseal chondrocyte apoptosis. Size of the SOC was inhibited either pharmacologically employing axitinib (A–D) or genetically employing activation of *Gnas* (E, F). (A–B) Representative histology images of the tibia epiphysis (A) and quantification of the SOC size (B) of wild type mice injected with axitinib at various time periods. P refers to postnatal day. (C–D) Representative images (C) and quantification of TUNEL staining (green) (D) in the hypertrophic cells of the tibia growth plate of the axitinib injected mice. (E–F) Quantification of SOC size (E) and TUNEL staining (F) of the hypertrophic cells in the growth plates of $_{stop}Gnas^{R201H}$ (Control) and *Prx-Cre: Gnas^{R201H}* (*Gnas-Tg*) mice. DAPI was used for counterstaining (blue) in (C). Data are means ± SD, two-tailed Student's t-test. In (A–F), n = 3. ns, not significant.

Summarizing, both pharmacological and genetic manipulation of the SOC confirmed its protective effect on hypertrophic chondrocytes, which is in agreement with the results obtained from ex vivo models and mathematical predictions.

## Mechanical properties of hypertrophic chondrocytes make them vulnerable to the mechanical stress

In another mouse model examined, where the formation of the SOC is delayed substantially by chondrocyte-specific ablation of the salt-induced kinase 3 (Sik3) (Col2a1-Cre:Sik3$^{F/F}$ mice) (*Sasagawa et al., 2012*), we noticed increased death of the hypertrophic chondrocytes not throughout the hypertrophic zone, but mostly at the sides of the growth plate (*Figure 9A–B*). In this strain, hypertrophy is slightly delayed and we thought this observation might reflect a phenomenon overlooked in our previous experiments, that is, the higher load at the edges or its different distribution during locomotion. To explore this further, we applied loading to our SOC ± model at an angle from the lateral side and examined the distribution of the cell death along the medial-lateral axis within the growth plate (*Figure 9C*). We noticed that chondrocytes on the side opposite to the loading underwent more extensive cell death (*Figure 9D–E*). This uneven distribution of cell death was also reflected in the higher number of TUNEL-positive cells and levels of YAP at the side opposite to the one where angled loading was applied (*Figure 9—figure supplement 1C,E,G,I*). Vertical loading did not affect the medial-lateral distribution of cell death, thereby serving as a control (*Figure 9—figure supplement 1A,B,D,F,H,J*).

This medial-lateral distribution of the cell death does not match the distribution of the octahedral stress, since upon angled loading this stress is low at the sides (*Figure 1F*). However, the distribution of the highest compressive principal stress and hydrostatic stress upon angled loading (*Figure 1—figure supplement 2*, see also *Figure 9F* and *Figure 9—figure supplement 1M* for physiological loading range) matches well with the observed distribution of cell death along the medial-lateral axis. Hydrostatic stress can be excluded since it increases upon vertical loading in the central part and at edges, and therefore does not match the distribution of cell death upon vertical load (*Figure 1—figure supplement 2*, *Figure 9—figure supplement 1M*). Thus, we concluded that directional compressive (principle) stress appears to be the one that overlaps well with the spatial distribution of cell death under various loadings, and is likely, together with octahedral stress, harmful for hypertrophic chondrocytes. Vulnerability to compressive stress suggests that hypertrophic chondrocytes have a very low Young's modulus (i.e., the ratio of stress to strain). To check this directly, we utilized atomic force microscopy (AFM) to measure cell stiffness in live tissue sections as described (*Xu et al., 2016*). According to the mathematical prediction, we found that hypertrophic chondrocytes were only approximately 25% as stiff as the columnar chondrocytes from which they are derived (*Figure 9G–H*).

Therefore, it seems plausible that this relatively low stiffness of hypertrophic chondrocytes, in combination with their large size, render them particularly vulnerable to mechanical loading.

## The involvement of mechanical versus genetic factors in the development of the SOC

As mentioned in the introduction, many investigators presently believe that the formation of the SOC is stimulated by mechanical factors (*Carter and Wong, 1988*; *Chagin et al., 2010*; *Klein-Nulend et al., 1986*; *Stevens et al., 1999*; *Sundaramurthy and Mao, 2006*; *Wong and Carter, 1990*). On the other hand, the observation that the thumb and fingers on the same wing of Chiroptera ossify at different times in utero (*Figure 2*) indicates that genetic factors may play a more important role in this formation. To explore this proposal, we examined the effects of unloading during limb development in rats and characterized ossification in a newt species during its terrestrial stage of growth.

On postnatal day 10, the hindlimbs of rats were unloaded either by resection of the sciatic nerve or fixation with kinesiology tape (which resembles fixation with a cast but is lighter and easier to change daily). Analysis at 23 days of age revealed that neither of these forms of unloading had affected the size of the SOC (*Figure 10A–D,F–I*), although in both cases bone mass was reduced dramatically (*Figure 10E,J*). The inability of the animals to use their nerve-resected leg was confirmed by testing footprint behavior (*de Medinaceli et al., 1982*; *Figure 10—figure supplement 1*) and this leg was also insensitive to pricking with a needle (video-recordings made every 3–4 days are available upon request). The inability to use the taped leg was also documented by video-recording every 3–4 days (available upon request).

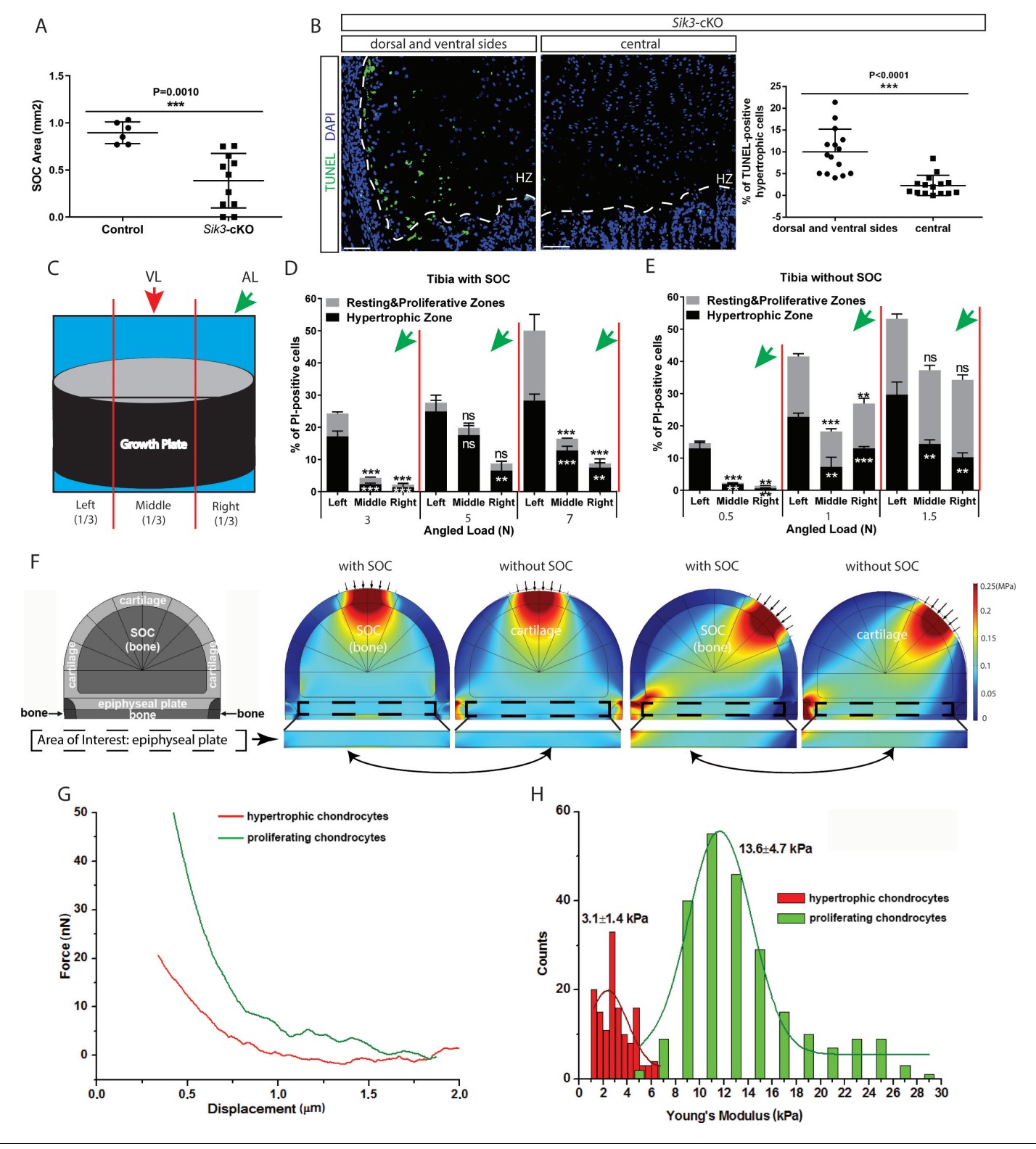

**Figure 9.** Low stiffness of hypertrophic chondrocytes renders them sensitive to mechanical stress. (**A**) Quantification of the SOC size of postnatal day 28 *Sik3*[F/F] (Control) and *Col2a1-Cre:Sik3*[F/F] (*Sik3*-cKO) mice. (**B**) Representative images and quantification of TUNEL staining (green) between the lateral and middle parts of *Sik3*-cKO mice. DAPI was used for counterstaining (blue). HZ, hypertrophic zone. In (**A–B**), data are means ± SD, two-tailed Student's t-test, n = 3 for 'Control' and n = 5 animals for '*Sik3*-cKO' whereas each dot represents one histological section analyzed. (**C**) Schematic illustration of a section plan and loading directions in relation to the quantitative analysis of cell death distribution in the lateral-medial direction

*Figure 9 continued on next page*

*Figure 9 continued*

presented in (D–E). The growth plate is divided into three equal parts, (left, middle, and right) in relationship to the angle at which the load is applied (green or red arrows). (D–E) Quantification of propidium iodide (PI)-positive chondrocytes in the resting and proliferative zones and hypertrophic zone of tibias with (D) and without (E) SOCs subjected to an angled load (AL, green arrow, always applied from the right). Data are means ± SD, one-way ANOVA. **, p<0.001 and ***, p<0.0001 in comparison with the left portion. ns, not significant. The black and white asterisks indicate the significance for the resting and proliferative and hypertrophic zones, respectively. (F) Principal compressive stress from FEA modeling with physiological loading level. The small arrows indicate the direction of loading. The two-headed curved arrows indicate areas for comparison. (G) Typical force curves for hypertrophic and proliferating chondrocytes obtained by atomic force microscopy. (H) Comparison of the elastic moduli of hypertrophic and proliferating chondrocytes.

The online version of this article includes the following figure supplement(s) for figure 9:

**Figure supplement 1.** The lateral-medial distribution of cell death upon angular loading.

These experiments indicate that the development of the SOC is primarily under genetic control. To obtain further support for this conclusion, we analyzed the epiphysis in the salamander *Notophthalmus viridescens*, an anamniote that do not normally develop a SOC. However, these animals can be raised under terrestrial conditions for several months. Therefore, we raised the newt *Notophthalmus viridescens* first under aquatic conditions, thereafter under terrestrial conditions (the eft stage, characterized by much slower growth than during the aquatic stages [*Hurlbert, 1969*]) and then under aquatic conditions again. No SOC was formed during terrestrial growth (*Figure 10—figure supplement 2A*), although calcification of cartilage during this period could be detected by Von Kossa staining (*Figure 10—figure supplement 2A*, middle panel). The numbers of apoptotic (*Figure 10—figure supplement 2B*) and proliferating chondrocytes (*Figure 10—figure supplement 2C*) were not dramatically different under aquatic and terrestrial conditions.

Altogether, these observations indicate that development of the SOC is governed primarily by genetic mechanisms designed to meet future mechanical demands, which aligns well with the evolutionary and zoological observations.

## Discussion

We show here that the SOC protects hypertrophic chondrocytes from mechanical stress and likely appeared in evolution in association with this function. We found that hypertrophic chondrocytes are characterized by very low mechanical stiffness and high vulnerability to loading. It is likely that their low mechanical stiffness in combination with the large size makes them vulnerable to various mechanical stresses, including the octahedral and principle compressive stresses. These mechanical vulnerabilities may be intrinsic to the process of chondrocyte hypertrophy per se, which is characterized by an enormous and rapid increase in the cellular volume, that is, twentyfold within 12 hr (*Cooper et al., 2013*). At the same time, chondrocyte hypertrophy is a key part of the process of endochondral bone formation, not only contributing 59% of longitudinal growth (73% if the corresponding extracellular matrix is taken into account [*Breur et al., 1994*; *Wilsman et al., 1996*]), but also coupling cartilage resorption to the formation of the underlying bone tissue (e.g., via secretion of Ihh and VEGF, as well as matrix calcification) (*Kozhemyakina et al., 2015*). The growth of long bones via hypertrophic intermediates (i.e. endochondral bone formation) has been evolutionarily conserved for at least 400 ± 20 Mya (*Sanchez et al., 2014*; *Sanchez et al., 2016*) and it seems plausible that early in evolution when growth occurred in the aquatic environment, no specific mechanical demands were posed on the hypertrophic chondrocytes due to compensation of body mass/ground reaction force by buoyant force. However, the transition to a fully terrestrial environment, unequivocally coupled with the loss of the buoyant force, places new mechanical demands on the growing skeleton, such as a need to withstand the body mass, and required additional adaptations of the skeleton.

The observation that longitudinal growth is sensitive to mechanical loads was documented almost two centuries ago and formulated during the period of 1862–1897 as the Hueter-Volkmann law, which states that '*increased mechanical compression reduces longitudinal bone growth while reduced mechanical loading increases it*' (*Mehlman et al., 1997*). This law was confirmed by numerous subsequent studies. For example, *Matsuda et al., 1986* revealed that the growth of the tarsometatarsus of roosters was delayed when exposed to a strenuous exercise. This observation was

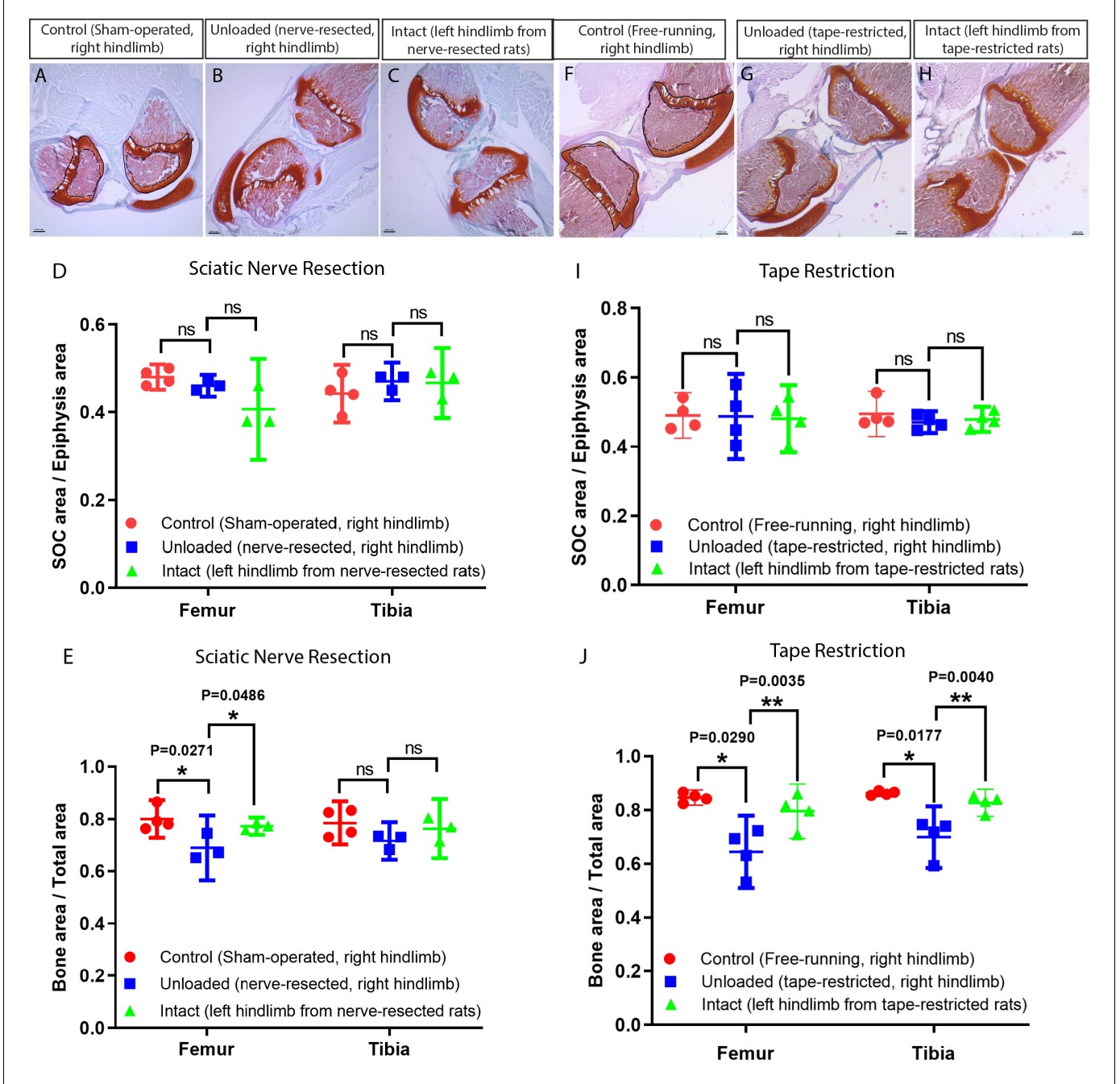

**Figure 10.** Unloading of the hindlimbs in growing rats by either sciatic nerve resection or by tape restriction. Sciatic nerve resection (A–E) or leg restriction with a kinesiology tape (F–J) was done at 10 days of age and both hindlimbs (manipulated and intact) were analyzed at day 23. For (A–E) sham-operated littermates were used as a control (only right hindlimb was utilized), whereas for (F–J) unmanipulated littermates were used as a control (only right hindlimb was utilized). (A–C, F–H) Representative Safranin O and Fast Green stained images of the knee joint. Black dash lines outline the SOC and solid lines outline the epiphysis area in A and F. Quantification of SOC size (normalized to total epiphysis size) (D, I) and bone mass (within the primary spongiosa region) (E, J). Bars are means ±95% confidential interval and every dot represent an individual animal, two-tailed Student's t-test. ns, not significant.

The online version of this article includes the following figure supplement(s) for figure 10:

**Figure supplement 1.** Footprint analysis revealed a clear difference between the right hindlimb of the sciatic nerve-resected and the sham-operated rats.

**Figure supplement 2.** Cartilage calcification but absence of the SOC formation in the newt growing on land.

further extended by *Reich et al., 2005*, who showed that chicks freely running with backpacks loaded with 10% of their body mass experience growth retardation associated with increased resorption of the epiphyseal cartilage. *Kiiskinen, 1977* showed that the longitudinal growth of the femur of young mice was retarded after several weeks of intensive training. Our data suggest that such a sensitivity of the growing bones to excessive load is due to the mechanical properties of hypertrophic chondrocytes and their vulnerability to mechanical stresses. It is likely that the cell death observed in our ex vivo and in vivo experiments is rather a far extreme manifestation of the cellular stress experienced by hypertrophic chondrocytes. The lower levels of cellular stress might have lesserbut yet negative impact on the hypertrophy process, such as decreased production of extracellular matrix and signaling proteins, including the observed reduction in Ihh, PTHrP, and ColX expression. Supporting, decreased expression of ColX was observed in the chicks with backpacks (*Reich et al., 2005*).

Locomotion is thought to impose the most frequent and severe loads on limb bones in terrestrial tetrapods (*Biewener, 1990*; *Butcher et al., 2008*). In children, normal gait pattern generates peak loads of four to six fold of the body weight on the femur (*Burdett, 1982*; *Yadav et al., 2016*). Such loads would be expected to be even higher during intensive physical activity, like sprinting or jumping (*Burdett, 1982*; *Ozgüven and Berme, 1988*; *Perttunen et al., 2000*). Thus, the load of 0.2N used in the present study is within the physiological range for animals with 20-gram body weight and would impair hypertrophic chondrocytes if they are not protected. Therefore, it is plausible to assume that if mice would not develop the SOC, hypertrophy of their chondrocytes would be negatively impacted during locomotion, which is in line with our pharmacological and genetic manipulations where mice with under-developed SOC display increased death of hypertrophic chondrocytes. At the same time, our data show that the SOC provides very effective protection, increasing withstanding loads up to twenty-five-fold and likely allowing hypertrophic chondrocytes to bear the corresponding body weight under various locomotion modalities, including sprinting and jumping.

However, in absolute numbers, the loads in elephants and mice look far beyond any compensation achievable by the SOC. How does it work? Considering species with various body masses along the phylogenetic tree, it was first noted by Galileo Galilei (*Galile, 1974*) that the skeletal elements become more robust relative to the body mass (BM) as the latter increases. The relation of the skeletal elements to BM is allometric (i.e. adaptive) and is generally described by the power law equation $y = kx^a$, where $k$ represents coefficient and $a$ represents scaling factor. The highest correlation has been shown between stylopodial (femur or humerus) diameter (D) and body mass (*Campione and Evans, 2012*). However, the exact scaling factor varies among studies, depending on the type of species and taxa utilized, weight-range selection, number of outliers, statistical strategy, sample size, etc. For example, for ungulates, this relationship is calculated as $D=BM^{0.366}$ (*Anderson et al., 1985*), whereas for insectivores+rodents it is $D=BM^{0.3944}$ (*Bou et al., 1987*). The most comprehensive analysis of quadrupedal terrestrial tetrapods (including both mammals and reptiles) resulted in the equation $D = 1.26*(BM)^{0.364}$ (*Campione and Evans, 2012*). This allometric increase in bone diameter substantially reduces the pressure (load per area) on the growth plate with increased bone mass. For example, due to this allometry, the increase in body size by thousandfold elevates the pressure on the growth plate by 4.3–6.6 fold (depending on which scaling factor from above is employed). Thus, allometric scaling of the skeleton compensates for most of the mechanical stress associated with increased body mass in terrestrial tetrapods. However, this compensation is not complete and pressure on the growth plate gradually increases for heavier animals. Why this compensation is not complete is currently unclear but may be coupled to high energy demands or unbearable skeletal mass required for compensation exclusively via skeletal allometry. Indeed, simple calculations show that complete pressure compensation on the skeleton can be achieved with the scaling factor 0.5 (i.e., $D=BM^{0.5}$), which would result in 3.4-fold increase in diameter of every weight-bearing skeletal structure of a 10 kg animal, such as a dog. Various alternative adaptations are employed by nature to allow the skeleton to withstand the increasing body mass and preserve the bone safety factor (the ratio of failure load to functional load), which is surprisingly stable from a mouse to an elephant and ranging from two to four (*Biewener, 1990*). The most significant adaptations include musculoskeletal composition, postures, and gait patterns (*Biewener, 1990*; *Biewener, 1989*). These adaptations, thought to evolve for the preservation of the skeletal safety factor, may unequivocally provide additional protection to the growth plate structure.

These various adaptations need to be considered from the evolutionary perspective, where various traits can evolve independently, develop in parallel or converge. With few sub-specialized exceptions, the entire Mammalia class can be characterized by the development of the SOC. In contrast, within Sauropsida clade, which is subdivided into two clades, Lepidosauria and Archosauria, limbed lepidosaurs (lizards and *Sphenodon*) exhibit a SOC, whereas Archosauria (crocodilians, chelonians, birds, and dinosaurs) exhibits ossified protrusions of bone marrow into epiphyseal cartilage (*Haines, 1938*; *Haines, 1942*; *Barreto et al., 1993*; *Horner et al., 2001*; *Figure 1E*). These protrusions are well-developed with calcified or ossified walls and penetrating deep into the epiphyseal cartilage, sequestering hypertrophic chondrocytes in between them in non-avian dinosaurs and birds, but very rudimental in chelonians and crocodilians (*Haines, 1938*; *Haines, 1942*; *Barreto et al., 1993*; *Horner et al., 2001*). According to our mathematical modeling, these extended protrusions are more efficient in protecting the zone of hypertrophic chondrocytes, whereas the SOC also reduces the deflection of the joint during locomotion (*Figure 1F*), which would facilitate more efficient transmission of the ground reaction force vector toward the body movement. Whether this difference has an evolutionary advantage remains to be explored, but the observation that extant birds develop SOCs only in their weight-bearing long bones (*Watanabe, 2018*) might suggest that improving the mechanical properties of the cartilaginous joint has an additional evolutionary advantage. Thus, it is plausible that the appearance of the SOC in weight-bearing long bones of birds represents an evolutionary convergence. At the same time, the appearance of the SOC in Mammalia and Lepidosauria and ossified protrusions in Archosauria can be considered as alternative parallel traits. Interestingly, recent genome sequencing of tuatara, the most ancient extant creature of Lepidosauria, revealed remarkable similarities in genome architecture with mammalian lineages (*Gemmell et al., 2020*).

Of course, the evolution of skeletal traits should not be considered alone, but in a connection with other related traits, such as postures and gaits. In this context, Mammalia can be characterized by an erect posture with (nearly) parasagittal limb kinematics in which the limbs support the entire weight of the trunk (*Biewener et al., 1983*; *Biewener et al., 1988*; *Blob and Biewener, 1999*). Same as mammals, birds, and dinosaurs are characterized by an erect posture (*Carrano and Biewener, 1999*; *Alexander, 1985*) whereas limbed lepidosaurs, crocodilians, and chelonians have a sprawling posture with laterally projected limbs (*Gans and Webb, 1997*). In such a configuration, the weight of the body is essentially distributed as shear stress in the midshaft of the stylopod, thereby loading the bone in torsion (*Butcher et al., 2008*; *Blob and Biewener, 1999*; *Ashley-Ross, 1994*; *Sanchez et al., 2010*). The hypertrophic chondrocytes are therefore not highly stressed in compression. Contrary to chelonians, some limbed lepidosaurs and crocodilians are capable of semi-erect posture (*Clemente et al., 2008*; *Schuett et al., 2009*). This is an intermediate configuration between the sprawling and erect posture (*Gans and Webb, 1997*). In that position, during the phases of fast locomotion, the limb-bone strain can significantly increase in axial compression and bending (*Blob and Biewener, 1999*; *Biewener, 2005*; *Clemente et al., 2011*; *Jd, 2002*). Crocodilians partition their habitat (to maximize feeding and/or minimize interference and predation), the juveniles are often restricted to shallow aquatic margins (*Jc, 1994*; *McNease, 1989*; *Tucker et al., 1997*). They barely go out of the water and only occasionally use a semi-erect posture. For these reasons, very little compressive stress is applied to their limb-bone growth plates at the juvenile stage. This may be the reason why crocodilians develop no SOC at their limb-bone extremities (*Haines, 1938*). On the contrary, all lizard limb-bones have SOCs (*Haines, 1938*; *Haines, 1942*; *Ml, 1884*; *Sanchez et al., 2008*) or calcified secondary center (SC) (in sphenodontids, *Figure 1D*; *Haines, 1939*). Lizards are terrestrial at the juvenile stage. In this exposed environment, they need to sprint fast and jump to avoid predation and hunt (*Dr, 1996*; *Marsh, 1988*; *Huey, 1982*; *Gj, 1985*). Many of them use a semi-erect or bipedal posture for this (*Gans and Webb, 1997*; *Clemente et al., 2008*; *Schuett et al., 2009*). Thus, the evolvement of the SOC may provide an advantage in terms of protection of the hypertrophic chondrocytes, simultaneously with improved firmness of the joint structure during the juvenile stage. Tortoises also develop exclusively on land. However, they only display this lateral limb projection typical of sprawling posture, which induces shear stress on the femoral midshaft (*Butcher and Blob, 2008*). No compression has been reported so far in the limb bones of tortoises. In addition, they move in such a slow manner (*Butcher et al., 2008*; *Butcher and Blob, 2008*; *Walker, 1971*; *Zani et al., 2005*) that the threshold of strain-related stimuli is too low (*Butcher et al., 2008*) to constrain the microanatomy of limb bones

(*Nakajima et al., 2014*) and stress their growth plates. This probably explains why SOCs are not present in terrestrial tortoise limb bones.

These evolutionary considerations lead us to propose that the development of SOCs (in mammals and Lepidosauria) or extended ossified protrusions (Archosauria) can be considered as parallel evolutionary traits evolved together with various postures for adaptation to the weight-bearing demands of terrestrial environments, including the mechanical stress placed on the epiphyseal cartilage in juveniles. At the same time, the evolutionary transition from water to land was associated with numerous other changes in the endocrine, respiratory, secretory, and other systems, most of which exert or may exert various effects on the skeleton. For example, the physiological functions of calcium influence the regulation of numerous types of bone cells. Similarly, the translocation of hematopoiesis from the kidney in fish to the bone marrow in terrestrial animals also affects bone physiology. Thus, mechanical adaptation may not be the only explanation for the development of the SOC.

From an evolutionary perspective, adaptive development of the SOC suggests an underlying genetic mechanism(s), as opposed to the widely held view that mechanical stress causes this development (*Carter and Wong, 1988*; *Klein-Nulend et al., 1986*; *Stevens et al., 1999*; *Sundaramurthy and Mao, 2006*; *Wong and Carter, 1990*). Indeed, in many instances, the SOC is formed during development before the application of mechanical forces. One well-known example of this is provided by newborn marsupials (e.g., the kangaroo), which are quite immature when born and need to use their forelimbs (which are much more ossified at this point than their hindlimbs) to climb into the pouch (*de Oliveira et al., 1998*). Another example involves ungulates, who must be able to walk and run within a few hours after birth. Accordingly, in calves, the SOC develops well before birth (*Geiger et al., 2014*; *Winters et al., 1942*). The heterochrony in the development of the SOC in different bones of jerboa and Chiroptera observed here is in line with these observations. Furthermore, such temporal variation even before birth and within the same limb of Chiroptera prepares adequately for future mechanical demands (see further above). Furthermore, early mechanical unloading of the hindlimbs of rats had no effect on the development of the SOC and raising *Notophthalmus viridescens*, which do not naturally develop a SOC, under terrestrial weight-bearing environment did not promote the formation of this structure (this study and *Libbin et al., 1989*). All these findings indicate strongly that the development of the SOC is genetically programmed.

The nature of this genetic programming remains to be elucidated. Interestingly, it was reported recently that a single nucleotide neomorphic mutation in miRNA140 delays the appearance of the SOC substantially, without dramatically influencing chondrocyte hypertrophy (*Grigelioniene et al., 2019*), suggesting that the underlying genetic program can be fine-tuned relatively easy. Among potential endocrine factors, thyroid hormone (TH), which regulates skeletal development in all vertebrates (*Bassett and Williams, 2016*), promotes the development of the SOC. In TSHR-deficient mice formation of the SOC is delayed substantially and this effect can be reversed by T3/T4 (*Xing et al., 2014*). Moreover, inactivation of thyroid hormone receptor - alpha (THR1a) in chondrocytes impairs their hypertrophy, formation of the SOC, and overall bone growth (*Desjardin et al., 2014*). At the same time, the SOC develops at different time-points in different bones suggesting the involvement of local mechanism(s). Indeed, sophisticated pre-receptor control of the activity/availability of TH (e.g., through the tissue-specific expression of deiodinase enzymes and transporters) and of their receptors (e.g., through the tissue-specific expression of these receptors and their co-activators and co-repressors) allows tight regulation of the timing and location of responsiveness to this hormone (*Williams, 2013*). Since TH influences the expression of approximately 400 early-response genes in the growth plate (*Desjardin et al., 2014*) and roughly the same number of genes during metamorphosis in amphibia (*Das et al., 2009*), it is difficult at present to even speculate about which specific genetic program triggered by TH might be involved in regulating the development of the SOC.

In conclusion, various independent approaches employed in this study (i.e., evolutionary analysis, mathematical modeling, biophysical tests, comparative zoology, biological experiments) all suggest that one evolutionary reason for the appearance of a spatially separated growth plate is to shield the hypertrophic chondrocytes from the mechanical stress associated with weight-bearing and, accordingly, preserve the mechanism of endochondral bone formation in terrestrial animals. Conceptual understanding of the underlying evolutionary, mechanical, and biological reasons behind the spatial separation of articular cartilage and the growth plate will provide a deeper understanding

of skeletal biology and evolution. This, for example, can improve the design of sporting activities for children or influence surgical interventions related to epiphyseal fractures.

# Materials and methods

**Key resources table**

| Reagent type (species) or resource | Designation | Source or reference | Identifiers | Additional information |
|---|---|---|---|---|
| Strain, strain background (*Mus musculus*) | C57BL/6 (mouse strain) | Charles River Laboratories | | |
| Strain, strain background (*Rattus norvegicus*) | Sprague Dawley (rat strain) | Janvier Labs | | |
| Strain, strain background (*Mus musculus*) | *Col2a1-Cre:Sik3$^{F/F}$* (mouse strain) | Dr. Henry Kronenberg, Massachusetts General Hospital | | |
| Strain, strain background (*Mus musculus*) | *Sik3$^{F/F}$* (mouse strain) | EUCOMM | | |
| Strain, strain background (*Mus musculus*) | *Col2a1-Cre* (mouse strain) | Richard R. Behringer, University of Texas MD Anderson Cancer Center | | |
| Strain, strain background (*Mus musculus*) | *Prx-Cre:Gnas$^{R201H}$* (mouse strain) | Dr. Murat Bastepe, Massachusetts General Hospital | | |
| Antibody | cleaved caspase 3 (rabbit polyclonal) | Cell Signaling | #9661 | 1:500 |
| Antibody | YAP (mouse monoclonal) | Santa Cruz | sc-101199 | 1:50 |
| Antibody | p73 (rabbit polyclonal) | Abcam | ab137797 | 1:100 |
| Antibody | ki67 (rabbit monoclonal) | Invitrogen | MA5-14520 | 1:20 |
| Antibody | MCM2 (rabbit polyclonal) | Abcam | ab4461 | 1:250 |

## Synchrotron scanning and 3D modeling

All these scans were performed at the beamline ID19 (European Synchrotron Radiation Facility, France) with a current of 200 mA or 16 bunch. Scanning and 3D-modeling of the humerus of *Eusthenopteron* were performed as described by *Sanchez et al., 2014*. The data were reconstructed using a single distance phase retrieval approach (*Sanchez et al., 2012*) based on a modified version of the algorithm of *Paganin et al., 2002*, applying an unsharp mask to the radiographs after the phase retrieval to enhance the trabecular mesh. The humerus of *Seymouria* was imaged with a voxel size of 3.48 µm using an optical system associated with a 47 µm-thick GGG (i.e., a gadolinium gallium garnet crystal) scintillator and a FreLON 2k14 CCD detector (*Labiche et al., 2007*). The propagation distance was of 700 mm. The gap of the W150m wiggler was opened to 37 mm. The sample was imaged at 80 keV using a beam filtered with 0.25 mm of tungsten and 2 mm of aluminum. In half-acquisition conditions, 4998 projections were produced over 360° with a time of exposure of 0.15 s. Binned images (final 6.96 µm voxel size) were reconstructed, segmented, and 3D-modeled in the same manner as *Eusthenopteron* (*Sanchez et al., 2014*). The scan of the humerus of *Sciurus vulgaris* was made with the same voxel size (3.48 µm) as for the humerus of *Seymouria* using the same optics (FreLON 2k14 CCD camera and a 47µm-thick GGG scintillator). The sample was imaged with a propagation distance of 500 mm. The beam was filtered with 0.14 mm of tungsten, 0.14 mm of copper, and 2.8 mm of aluminum. The gap of the W150m wiggler was opened to 50 mm, which resulted in energy of 65.8keV. In half-acquisition conditions, 4998 projections were produced over 360° with a time of exposure of 0.15 s.

## MicroCT scan of *Pipistrellus pipistrellus s.l*

The 1–3 day-old bat was scanned in Vienna with a SkyScan 1174 as previously described (*Metscher, 2009*).

## Mice and rats

30-day-old C57BL/6 mice (body weight 14.59 ± 0.72 g) and 10-day-old Sprague Dawley rats (body weight 20 ± 0.49 g) were purchased from Charles River Laboratories and Janvier Labs (Europe). C57BL/6 mice used for the axitinib experiments were purchased from Charles River Laboratories. *Col2a1-Cre:Sik3$^{F/F}$* mice were obtained from Dr. Henry Kronenberg, Massachusetts General Hospital. The *Sik3$^{F/F}$* mice were originally purchased from EUCOMM and the *Col2a1-Cre* mice were obtained from Richard R. Behringer, University of Texas MD Anderson Cancer Center (*Ovchinnikov et al., 2000*). *Prx-Cre:Gnas$^{R201H}$* mice were generated by Dr. Murat Bastepe, Massachusetts General Hospital and were described previously (*Karaca et al., 2018*).

## Dissection, loading, and culturing of the tibia and femur

From both hind legs of 30-day-old mice and 10-day-old rats, the tibia and femur bones were dissected out aseptically in DMEM/F12 medium supplemented with 50 µg/mL gentamycin (dissection medium) on ice, with the removal of as much surrounding tissue as possible. The height and diameters of the top and side surfaces of the epiphysis were measured with a digital caliper. For each animal, one set of tibia and femur was used for control and the other for treatment. All comparisons involved the bones from the same animal.

Still, in dissection medium on ice, the bones were subjected to pressure with the Instron ElectroPuls E1000 Test Instrument. The load was applied at a speed of 0.1 N per second and load/digital position curves recorded in the Bluehill three software. Control bones went through the same procedures, but without loading.

For culturing, bones were first washed twice in dissection medium before being placed into DMEM/F12 medium supplemented with 1 mM β-glycerophosphate, 0.2% bovine serum albumin, 50 µg/mL ascorbic acid, and 50 µg/mL gentamycin for 48 hr as described previously (*Cooper, 2009*). Propidium iodide (stock solution at 1 mg/mL, Sigma) was added to the culture medium at a ratio of 1:50 30 min before termination of culturing. The tibia and femur were then fixed in 4% PFA overnight, decalcified, cleared in 30% sucrose overnight (at 4˚ C), and finally embedded in optimal cutting temperature (OCT) compound for frozen sectioning. When applying the load, the medial tibia plateau flanked by the tibia bone grooves is always facing one direction and the angled pressure was applied always to one side of the plateau (on top of one of the grooves). When embedding in OCT for sectioning, the bone is always placed into the grid with the medial tibia plateau facing upward and sectioned frontally to ensure the pressure direction during imaging.

## Calculation of mechanical and material properties

The determination of the position of the bone relative to the position of its fixture yielded the deformation value. Stiffness was defined as the slope of the initial linear portion of the load-deformation curve. Stress was calculated by distributing the load equally over the top (vertical load) or side surface (angular load) of the epiphysis, assuming that both of these surfaces were oval. The strain was calculated by distributing the bone deformation equally along the height of the epiphysis. The initial linear portion of the stress-strain curve up to 0.2% offset was plotted together with the stress-strain curve itself in the same graph and the intersection of the line and curve defined as the 0.2% offset yield strength.

## Axitinib injection

Axitinib (Sigma) was dissolved in DMSO and injected intraperitoneally at a dose of 0.25 mg per animal per day for the C57BL/6 mice for 8 (P21 to P27) or 10 days (P18 to P27 or P21 to P30). The control group of the C57BL/6 mice received the same volume of DMSO injection each day. Transgenic *Prx-Cre:Gnas$^{R201H}$* mice and corresponding *Gnas$^{R201H}$* controls were injected daily 0.017 mg per gram of body weight between postnatal 21 and 29 days of age to block the elimination of dead hypertrophic chondrocytes by ingrowing blood vessels. Animals were sacrificed one day after the last axitinib injection.

## TUNEL staining

About 30 µm thick sections of cultured tibia bones were treated with proteinase K (Ambion) at 10 µg/mL for 40 min at 37° C before applying the TUNEL reaction mix (Roche Inc) for 90 min. The cells were then counter-stained with DAPI for 5 min.

## Quantification of positive cells (PI+, TUNEL+, YAP+, etc)

For each data point of the PI+ and TUNEL+ cell number quantification, at least three sections per animal and at least three animals were analyzed. Double-blind counting was performed for all quantification analysis, including the mediolateral distribution of PI+, TUNEL+, caspase-3+, p73+, YAP+, and double p73+YAP+ cells.

## Immunofluorescent staining

30 µm sections of cultured tibia bones were blocked in 3% normal horse serum for 1 hr, followed by overnight incubation with primary antibodies against cleaved caspase-3 (1:500) (Cell Signaling), YAP (1:50) (Santa Cruz), p73 (1:100) (Abcam), Ki67 (1:20) (Invitrogen), or MCM2 (1:250) (Abcam) at 4° C. Secondary antibodies tagged with fluorophore were then added for one hour followed by DAPI counter-staining for 5 min.

## In situ hybridization

In situ hybridization was performed on 30 µm sections as described previously (*Chagin et al., 2014*) utilizing a probe against ColXa1 (a gift from Prof. Bjorn Olsen, Harvard Medical School) labeled with digoxigenin (DIG) in accordance with the manufacturer's instructions (Roche Inc). Samples were then treated with either DIG antibody Fab fragments (Sigma) and NTM (Sigma) for colorimetric visualization or DIG antibody HRP (Sigma) and TSA kit (Perkin Elmer) for fluorescent visualization.

## Atomic force microscopy

About 120–150 µm thick sections of live rat tibia bones were obtained by vibratome (Thermo Fisher Scientific) and mounted onto glass discs using silicone grease from the Bruker fluid cell accessory kit (Bruker). The force spectroscopy measurements were performed using a MultiMode eight atomic force microscope with a Nanoscope V controller and E scanner (Bruker). The regions of cells of interest for the acquisition of force-distance curves were selected under the optical microscope in combination with the AFM instrument.

The force-distance curves were acquired employing CP-PNP-BSG colloidal probes (NanoandMore GmbH, Germany) with a 5 µm borosilicate glass microsphere attached to the 200 µm cantilever. The spring constants of the cantilevers (measured by the thermal tune procedure) were 0.06–0.09 N/m.

All measurements were conducted at 25°C and all tissue was handled in DMEM/F12 HEPES-containing medium on ice. At least 70 individual force-distance curves were acquired for each type of cell by ramping over the surface and a total of 30 cells were measured from three different animals. These force-distance curves were processed with the NanoScope Analysis v.1.10 software (Bruker). Utilizing retract curves, the elastic modulus E was extracted from these force-distance curves by fitting in accordance with the Hertzian model of contact mechanics.

## Nanoindentation

Local mechanical characteristics of 10-day-old rat and 30-day-old mice cartilage samples were measured with a Piuma Nanoindenter (Optics11, Netherlands) adapted for soft materials. Measurements were performed on 220–250 µm sections of live tissues (without fixation and freezing) obtained by vibratome (Thermo Fisher Scientific) at 2-4°C. After the measurement, the sections were stained with PI (Sigma) and CalceinAM (Sigma) to confirm cell viability.

Nanoindenter includes a controller, an optical fiber, and a spherical tip for the force-displacement curves acquisition. The tip is attached to a flexible cantilever, the displacement of which after the contact with a surface is measured using an interferometer via an optical fiber.

To measure the Young's modulus, the probe was immersed for 5 µm into the sample at each point of measurement. The Young's modulus for each point was computed according to the Hertzian contact mechanics model for a spherical body indenting a flat surface, using the built-in Piuma software.

For the study of 10-day-old rat and 30-day-old mice cartilage mechanical characteristics, we used a cantilever with the spring constant of 4.14 N/m and a tip with the 45.5 µm radius of curvature. The measurements were conducted in phosphate buffered saline (PBS) cooled to 2–4℃ to maintain cell viability. The samples were immobilized with a specially designed holder immersed in cold PBS. During the measurements, the probe was always located inside the fluid medium at a sufficient depth, in order to avoid measurement errors due to adhesion forces at the air-water boundary. The area of the Young's modulus mapping for rat samples was 1350 × 400 µm with the step of 45 µm by the X-axis and 100 µm by the Y-axis. For mice, the area of mapping was 585 × 360 µm with the step of 45 µm by the X- and Y-axes. Based on the results of the measurements, the effective Young's modulus was computed, its distribution over the surface was plotted and mean ± standard deviation (SD) Young's modulus value calculated.

For the experiment assessing the effect of axitinib on the mechanical properties of epiphyseal cartilage, between 14 and 96 measurements of cartilage stiffness with nano-indenter were performed for each animal and the median (due to not normal distribution of the measurements) was chosen as the closest 'true' value representing cartilage stiffness for each mouse (six controls and seven treated mice). These values were employed to assess the effect of the treatment, their normal distribution verified by the Shapiro–Wilk test (passed; p=0.4051 and p=0.3921 for control and axitinib groups, respectively) and equality of distributions compared by unpaired t-test with n = 6 for control and n = 7 for treated.

## Finite element analysis

The effects of a SOC on the distribution of stress in the zones of bone growth were explored by numerical simulations. Following the reasoning of *Carter et al., 1998*, a plane-strain 2D domain was considered, to simplify the complex geometry of the bone end, and thereby clarify fundamental aspects of the response. The model thus considered five sub-domains. The geometry of this simulation domain was a slight modification of *Carter et al., 1998*, with a total height of 66 mm, the radius at the top of 17 mm, and bottom width of 25 mm. The cortical bone was given a width of 2.5 mm (measured horizontally). The thickness normal to the 2D plane was by definition 1 mm. Only the top part of the model is shown in the results figures.

The main parameter in these simulations was the presence or absence of a SOC or other stiffer tissues, introduced by providing this domain with stiffness at least equal to that of the cartilage domain. Non-linear strains were taken into consideration, but the materials in all subdomains were considered to be linearly elastic, as described by a Young's modulus $E_i$ and a Poisson ratio $\nu_i$. We used the same material properties as (*Carter et al., 1998*), and in particular for cartilage $E_C = 6$ MPa, $\nu_C = 0.47$. When included, the SOC was defined in a subdomain with $E_D = 500$ MPa, $\nu_D = 0.20$. Thus, the stiff tissue had the material properties of dense cancellous bone, that is, approximately eighty-fold stiffness compared to cartilage.

In all cases, the bottom edge of the complete model had zero displacements vertically, and the outside edges of the cortical bone zero displacements horizontally. Applied to different 45° sectors of the semi-circular top, the pressure was modeled as maximal on the center of the given sector (300 kPa for physiological level, 3 MPa for supra-physiological), falling quadratically and symmetrically to zero on the two neighboring sector borders. In light of the symmetry of the domain, only the top- or right-hand side was loaded (except for a verification case to *Carter et al., 1998*, not shown). The direction and integrated total force of loading were affected slightly by the finite deformation of the domain.

Different components or comparison values of stress (measured as the second Piola-Kirchhoff stress relative to the unstressed reference volume) were evaluated. The primary focus was on hydrostatic stress, the lowest principal stress (always compressive), and the octahedral shear stress, providing different perspectives on the response to external loading. Since the situations considered were dominated by compressive stresses, the signs for the two first have been changed in figures. Stress values are plotted on the deformed shape of the domain, without magnification of the deformation.

Numerical simulations of the response to loading were based on parameterized finite element approximations and performed in the Comsol Multiphysics software (version 5.2, Comsol AB, Stockholm, Sweden).

With similar loads, this model resulted in stress distribution in and deformation of the cartilaginous epiphysis closely similar to those reported by *Carter et al., 1998*.

Model design for stem tetrapods and archosaurs were based on *Sanchez et al., 2014*; *Sanchez et al., 2016* and (*Chinsamy and Abdala, 2008*; *Ray and Chinsamy, 2004*), respectively.

## Quantitative PCR

Proximal end growth plate of mouse and rat tibia was dissected and homogenized in liquid nitrogen for subsequent RNA extraction by Trizol and RNA purification by the RNeasy kit (Qiagen). Only RNA samples with an A260/A280 value between 1.8 and 2.0 were used. Extracted RNA was reverse transcribed into cDNA using a kit from Takara. qPCR was performed using the SYBR Green I supermix (BioRad) and analyzed using the ΔCt method by normalizing with the housekepping gene GAPDH. Three growth plates from independent animals were analyzed for each condition.

## Sciatic nerve resection

Wistar-Kyoto rats at P10 were anesthetized with 20 mg/kg Zoletil (Vibrac). On the posterior side of the right leg, the skin and muscle were dissected, and the sciatic nerve was isolated. For experimental animals, a 5 mm nerve fragment was excised; for sham-operated animals, the nerve was returned to the soft tissue without resection. The skin at the operation site was sutured and, after recovery from anesthesia, the pups were returned to their mother. The suture was treated with povidone-iodine for 7 days after surgery. Foot sensitivity was tested at P16 using a needle. Video recordings of gaits were carried out at P11, P14, P20, and P22. The animals were sacrificed at P23 for tissue collection.

## Leg immobilization with kinesiology tape

The right legs of 10-day-old Wistar-Kyoto rats were immobilized using kinesiology tape (Bradex) and the left legs remain intact. Free-living animals were used as the control group. The tape was wrapped in two layers to fix the knee and ankle joint. As the rats grew, the tape was changed every 2 days. Rats' gait pattern was observed daily, and the video was recorded at P14, P20, and P22, to verify the unloading of the taped leg. Rats were sacrificed at P23 for tissue collection.

## SOC size evaluation

Knee samples were fixed in 3.7% formaldehyde for 1 day and decalcified in EDTA. About 20 µm cryosections were stained with Safranin O and Fast Green. SOC size was estimated in the middle of the epiphysis, identifiable by the presence of a cruciate ligament. SOC area was measured and normalized to the total epiphysis area, including the growth plate with Image J.

## Bone density evaluation

The primary spongiosa area (500 µm under the growth plate) on hematoxylin and eosin-stained sections was used to evaluate the bone density. The bone area that is stained with eosin was measured and normalized to the total primary spongiosa area with Image J.

## Footprint rat behavior test

The test was carried out on P20. The rat's hind legs were painted with iodine and the animal was allowed to move freely on a blank sheet of paper. After the rat walked along the sheet in one direction, the sheet was changed. For each rat, at least five records were made. Papers with footprints were scanned.

## Statistical analysis

The values presented as means ± standard deviations of at least three independent experiments. The unpaired Student's t-test and one-way ANOVA were utilized to calculate P-values if not otherwise indicated.

## Acknowledgements

This work was supported financially by EMBO Long-Term Fellowships (MX) and grants from the Swedish Research Council (ASC, SS, AI, AnSi), Karolinska Institutet (MX, ASC, LL, IA, including an SFO Stem/Regen junior grant to ASC), Bertil Hallsten Foundation (IA), EMBO Young Investigator

Program (IA), Åke Wiberg Foundation (IA), Chinese Scholarship Council (LL), Stiftelsen Frimurare Barnhuset i Stockholm (PN, MX), Museum of the Southern Jutland (PG), Olle Engkvist Stiftelse (AnSi), National Institutes of Health (MB: NIDDK RO1 DK073911; HK: NIDDK P1 DK001794). The Russian Science Foundation financially supported the experiments with atomic force microscopy (grant #18-15-00401 to PT) and mechanical loading experiments (grant #19-15-00241 to ASC).

O Lambert (IRSNB), MT Olsen (ZMUC), F Zachos (NMW), PHC Lina (Naturalis), D Kalthoff (Museum of Natural History, Stockholm, Sweden), and J Klembara (CU) provided access to specimens in the collections under their care. Synchrotron beamtime was allocated in response to a proposal accepted by the ESRF (EC203, SS; ES342, JE), as well as in-house (Paul Tafforeau). We are thankful to Prof. David Eilam (Tel-Aviv University) who kindly shared his opinion as well as a video-recording of jerboa gait pattern acquisition (*Figure 3—video 2*).

## Additional information

### Funding

| Funder | Author |
| --- | --- |
| EMBO | Meng Xie<br>Igor Adameyko |
| Vetenskapsrådet | Sophie Sanchez<br>Igor Adameyko<br>Andrei S Chagin<br>András Simon |
| Russian Science Foundation | Peter Timashev<br>Andrei S Chagin |
| Stiftelsen Frimurare Barnhuset i Stockholm | Meng Xie<br>Phillip T Newton |
| NIDDK | Murat Bastepe<br>Henry M Kronenberg |

The funders had no role in study design, data collection and interpretation, or the decision to submit the work for publication.

### Author contributions

Meng Xie, Data curation, Formal analysis, Validation, Investigation, Methodology, Writing - original draft, Writing - review and editing; Pavel Gol'din, Conceptualization, Investigation, Methodology, Cetaceans analysis; Anna Nele Herdina, Data curation, Methodology, Chiroptera analysis; Jordi Estefa, Ekaterina V Medvedeva, Lei Li, Phillip T Newton, Svetlana Kotova, Boris Shavkuta, Aditya Saxena, Lauren T Shumate, Brian D Metscher, Karl Großschmidt, Shigeki Nishimori, Anastasia Akovantseva, Data curation; Anna P Usanova, Anastasiia D Kurenkova, Anoop Kumar, Investigation; Irene Linares Arregui, Methodology; Paul Tafforeau, Kaj Fried, Mattias Carlström, András Simon, Christian Gasser, Murat Bastepe, Peter Timashev, Resources; Henry M Kronenberg, Kimberly L Cooper, Igor Adameyko, Resources, Writing - review and editing; Sophie Sanchez, Resources, Methodology, Writing - review and editing; Anders Eriksson, Conceptualization, Data curation, Methodology; Andrei S Chagin, Conceptualization, Resources, Supervision, Funding acquisition, Investigation, Methodology, Writing - original draft, Project administration, Writing - review and editing

### Author ORCIDs

Meng Xie (iD) https://orcid.org/0000-0002-6388-9789
Pavel Gol'din (iD) http://orcid.org/0000-0001-6118-1384
Brian D Metscher (iD) http://orcid.org/0000-0002-6514-4406
Paul Tafforeau (iD) https://orcid.org/0000-0002-5962-1683
András Simon (iD) http://orcid.org/0000-0002-1018-1891
Kimberly L Cooper (iD) http://orcid.org/0000-0001-5892-8838
Sophie Sanchez (iD) https://orcid.org/0000-0002-3611-6836

Igor Adameyko (iD) http://orcid.org/0000-0001-5471-0356
Andrei S Chagin (iD) https://orcid.org/0000-0002-2696-5850

## Ethics

Animal experimentation: all animal experiments were pre-approved by the Ethical Committee on Animal Experiments (N5/16, N187/15, 9091-2018, Stockholm North Committee/ Norra Djurförsökse-tiska Nämnden), the Institutional Animal Care and Use Committee of the Massachusetts General Hospital (Protocols #: 2005N000094 and 2004N000176) or the University of California San Diego (D16-00020) and conducted in accordance with the provisions and guidelines for animal experimen-tation formulated by the Swedish Animal Agency. Animal experiments involving limb unloading, AFM and nanoindentation were pre-approved by the Ethics Committee of the Sechenov First State Moscow Medical University (No. 07-17 from 13.09.2017, Moscow, Russia).

## Decision letter and Author response

Decision letter https://doi.org/10.7554/eLife.55212.sa1
Author response https://doi.org/10.7554/eLife.55212.sa2

## Additional files

### Supplementary files

• Source data 1. Numerical data for all graphs in the figures.

• Supplementary file 1. Primary data on the level of ossification of the individual bones of all species of Chiroptera analyzed.

• Transparent reporting form

### Data availability

All data generated or analysed during this study are included in the manuscript and supporting files.

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
