## [Decision Letter]

**Acceptance summary:**

The secondary ossification center is in support of articular cartilage of joints. It is important to understand the secondary ossification center is formed during development. Pathological changes in subchondral bone drives progression of joint arthritis. Your findings in this study provide useful information in understanding the pathogenesis of joint arthritis.

**Decision letter after peer review:**

Thank you for submitting your article "Secondary ossification center induces and protects growth plate structure" for consideration by *eLife*. Your article has been reviewed by three peer reviewers, and the evaluation has been overseen by a Reviewing Editor and Clifford Rosen as the Senior Editor.

The reviewers have discussed the reviews with one another and the Reviewing Editor has drafted this decision to help you prepare a revised submission.

Summary:

The paper analyses the role of the secondary ossification center (SOC) for the mechanical properties of long bones in support of growth plates by combining evolutionary developmental comparisons, mathematical modeling and functional analysis. The mechanical impact of SOC on the proliferation, and apoptosis of chondrocytes in the growth plate was investigated. The manuscript is well written and is educational to both evolutionary and bone development field.

Evolutionary comparison of long bone development and the analysis of load bearing bones at early postnatal stages strongly indicates that SOC development is related to life on land and to the mechanical loads the bones are exposed to. Mathematical modelling demonstrates that a SOC reduces the mechanical stress on the growth plate upon mechanical loading. Functionally, the authors investigate the mechanical stress upon loading in long bones with and without SOC and in mouse mutants with a delayed formation of the SOC. The loading leads to increases YAP-p73 induced apoptosis and that a SOC reduces the response significantly.

Overall, this manuscript is well written, interesting and will help to understand the role SOC during development and the interplay between articular cartilage and subchondral bone, and particularly in osteoarthritis. We have the following suggestions to improve the manuscript. We also realized that difficulties with the single mouse experiments, like the different developmental age of mice and rats in an evolutionary approach, which could be strengthened with the combination of the different mouse(rat) models (circumstantial evidence) and the evolutionary and mathematical analyses. Therefore, we would like to consider your revised manuscript within 4 months.

Essential revisions:

1) The lack of molecular mechanism that determines the presence or absence of the secondary ossification center. The authors' conclusions naturally lead to further questions such as:

a) Whether mechanical unloading prevents formation of secondary ossification.

b) How chondrocytes from animals with the secondary ossification respond differently to mechanical stimulation from those without it.

c) What inhibits/induces formation of the secondary ossification in animals that do/not have it during the development, etc.?

2) It is impressive that the investigators provide microanotomical characterization of SOC formation in different species of animals from 380-million-year-old lobe-finned fish to present mammals. However, when animals left the aquatic environment and began to live on land, the changes of mechanical loading only partially contribute to adaption of the skeletal system. For example, the changes in the endocrine system and the development of parathyroid gland (PTH becomes a hormone to regulate skeletal modeling and remodeling) may all play a role in the SOC formation. This question also applies to the comparative analysis of the cetaceans with the other two terrestrial animals.

3) The gait pattern of jerboa that showed in the video clip doesn't support that the jerboa only uses its front limbs for crawling during the first 2 weeks of age. Instead, they seem to use their hind limbs for crawling more.

4) There is a concern regarding the comparison between 10-day old rats and 30-day-old mice. The rodents are under fast growing during the first postnatal month. Although the indentation of cartilage of these two animal models is similar, the development of skeletal tissue such as muscle, ligaments may be different. Importantly, the intrinsic biological and/or mechanical properties, such as the proliferation rate or matrix protein synthesize capabilities of the chondrocytes in rat at P10 is likely distinctive to mouse at P30. In the Figure 5, it clearly shows that the cell number of the growth plate in the rat model in much more than the mouse model. Whether this characteristic affect the outcome?

5) The FEA simulation model is relatively simplified. For example, they assume both of the surfaces were oval and materials in all subdomains were considered to be linearly elastic, which brings concerns on the FEA to reflect the real mechanical situation.

6) The influence of genetic modulation or axitinib treatment on changes of phenotype cannot be excluded if no evidence shows that the biological properties of the chondrocytes do not change in the axitinib or Prx-Cre: Gsa model.

---

## [Author Response]

Essential revisions:1) The lack of molecular mechanism that determines the presence or absence of the secondary ossification center. The authors' conclusions naturally lead to further questions such as:a) Whether mechanical unloading prevents formation of secondary ossification.

To address this question unloading experiments should be done on animals before SOC appearance. Experimental animals that develop a SOC and are available to us (i.e., mice and rats) develop the SOC early in postnatal life, when pups are still dependent on their mother’s milk. This precludes us from performing classical unloading experiments such as tail suspension.

To overcome this issue, we have established and explored two models of hindlimb unloading in breastfeeding pups. One is sciatic nerve resection and another is a foot fixation by a kinesiology tape (resembling a cast fixation, but lighter in weight and exchangeable daily). The tape has a bitter taste and the mother does not remove it. Both experimental models were applied to rats at 10 days of age (prior SOC appearance) and analyzed at 23 days of age, when SOC is developed.

Unloading caused no differences in SOC size as compared with either sham-operated animals (separate animals were used for this control) or contralateral leg of the manipulated animals. At the same time clear decrease in bone mass was observed in the unloaded limb. Unloading models were verified as following: Inability to use the nerve-resected leg was confirmed by footprint behavior test, lack of innervation throughout the study period (one represented video is uploaded for review purposes). Tape-restricted unloading was confirmed by video-recording every 4 days (one represented video is uploaded for review purposes).

These new experiments are presented in new Figure 10 and Figure 10—figure supplement 1 (footprints) and reflected in the text (subsection “Mechanical properties of hypertrophic chondrocytes make them vulnerable to the mechanical stress”).

b) How chondrocytes from animals with the secondary ossification respond differently to mechanical stimulation from those without it.

Animals that develop and those which do not develop the SOC belong to different classes, e.g. Amphibia and Mammalia. We think direct comparison of such distinct animals as, for example, salamander and mice will be difficult to interpret in relation to mechanical stimulation specifically.

However, we have got an access to a salamander of *Notophthalmus* species, which can spend part of their juvenile life on land. Comparison of the epiphyseal cartilage from these animals raised in aquatic and terrestrial environment revealed no formation of the SOC while growing in land. However, we noticed partial calcification of epiphyseal cartilage during terrestrial period of growth. No differences were observed in the number of proliferative or hypertrophic chondrocytes.

These new data are presented in a new Figure 10—figure supplement 2 and reflected in the text (subsection “Mechanical properties of hypertrophic chondrocytes make them vulnerable to the mechanical stress”).

c) What inhibits/induces formation of the secondary ossification in animals that do/not have it during the development, etc.?

This is a very conceptual question since at present it is not even clear if genetic or mechanical mechanisms are the primary cause of SOC development. The view that mechanical stimulation drives SOC development is quite common in literature (Carter and Wong, 1988; Chagin et al., 2010; Kleinnulend et al., 1986; Stevens, Beaupre and Carter, 1999; Sundaramurthy and Mao, 2006; Wong and Carter, 1990) albeit our new data generated during revision (#1a and b above) strongly argue for genetic mechanism(s) underlying SOC development.

The general view is that the SOC develops similarly to the primary ossification center (POC), where chondrocyte hypertrophy and associated VEGF secretion attract invasion of blood vessels followed by osteo-progenitors (Maes et al., Developmental Cell 2010, 19:324-344). However, a recent study suggests that there might be some differences between POC and SOC formation. In the latter, mesenchymal cells of perichondrium migrate inside the epiphysis forming canals and secreting VEGF and thereby attracting vascular cells (Tong et al., Stem Cells 2019, 37:677-689). Still, what triggers the migration of perichondrial cells into the epiphysis is as yet unknown.

There are numerous genes, shown to be implicated in SOC development. These include:

– EGFR activity (Zhang X et al., J Biol Chem 2013, 8:3229-32240; ablation of EGFR in chondrocytes; slight delay in SOC development and also expanded hypertrophic zone, both are likely via downregulation of MMPs, no data on body size),

– Canonical Wnt signaling (Dao et al., JBMR 2012, 27:1680-1694; after activation of b-catenin in chondrocytes, analysis is done on E18.5, demonstrating accelerated hypertrophy in the epiphysis and metaphysis; notice that normally SOC starts to form in a mouse femur and tibia around P5),

– IGF-1 is claimed to be involved (Wang et al., ablation of IGF1R in osx-expressing cells slightly delays SOC formation and strongly decreases body size)

– Thyroid hormone promotes SOC development (Xing et al., 2014; the effect is vivid and reversible by addition of T3/T4 to TSHR deficient mice, hypertrophy is impaired in the absence of thyroid hormones). Generally, it is well known that thyroid hormone regulates ossification, predominantly via promoting chondrocyte hypertrophy. Inactivation of THR1a in chondrocytes impairs their hypertrophy, formation of the SOC and overall bone growth (Desjardin et al., 2014). Thyroid hormone changes around 400 early-response genes in the growth plate (Desjardin et al., 2014) and roughly the same number in amphibia when it induces metamorphosis (Das et al., 2009), making it difficult to do a qualified guess on the underlying mechanisms.

– Hypoxia pathway, where ablation of Vhlh in chondrocytes delays formation of the SOC (Pfander et al., Development 2004, 131:2497-2508) and ablation of Hif1a in chondrocytes causes cell loss in the area where SOC is supposed to be developed (Shipani et al., Genes and Development 2001, 15:2865-2876).

These and other sporadic observations suggest a close connection between SOC formation and chondrocyte hypertrophy within the epiphysis, and the majority of observations reflecting regulation of the SOC involve, in fact, either acceleration or inhibition of chondrocyte hypertrophy, cartilage resorption (i.e., metalloproteinases) or vascularization. In our manuscript we utilized this fact and to adjust SOC size via modulation of chondrocyte hypertrophy, employing mice with either SIK3 conditional ablation or Gsa conditional activation, or via modulation of vascularization.

However, the mechanism that triggers hypertrophy in a specific region of the epiphysis is not known. Furthermore, it is not known why it does happen in some epiphyses, but not others, like distal and proximal epiphyses of metatarsal bones. Interestingly, a recent study showed that neomorphic mutation in MIR140 (coding microRNA 140) causes a striking delay in SOC development without affecting hypertrophy (Grigelioniene et al., 2019). This point mutation changes a transcription regulation of numerous genes within epiphyseal cartilage as compared either with MIR140-deficient or wild-type mice and gene set enrichment analysis (GSEO) revealed impairment of the hypoxia pathway among others (Grigelioniene et al., 2019). Our brief evolutionary analysis revealed that species with and without SOC development do not differ in the MIR140 sequence, precluding us to consider this point mutation as a potential mechanism in evolutionary appearance of the SOC.

Summarizing all the above, the question raised by reviewers requires a deep and comprehensive study on its own, which, from our point of view, seems to be beyond the scope of the current manuscript. We have added two new paragraphs to the Discussion reflecting the most likely molecular mechanisms from the above and stressing that our data (particularly new data generated in response to the comments above) provide a general understanding that the process of SOC development is primarily regulated genetically but not mechanically, as it commonly seen.

2) It is impressive that the investigators provide microanotomical characterization of SOC formation in different species of animals from 380-million-year-old lobe-finned fish to present mammals. However, when animals left the aquatic environment and began to live on land, the changes of mechanical loading only partially contribute to adaption of the skeletal system. For example, the changes in the endocrine system and the development of parathyroid gland (PTH becomes a hormone to regulate skeletal modeling and remodeling) may all play a role in the SOC formation. This question also applies to the comparative analysis of the cetaceans with the other two terrestrial animals.

The reviewers are correct and transition from water to land was associated with numerous changes in the endocrine system and various other systems, such as respiratory, secretory, etc. Most of them have or may have systemic effects on the skeleton. For example, calcium physiology had wide consequences in the behavior and regulation of all bone cells. Similarly, the movement of hematopoiesis from the kidney in fish to the bone marrow in land animals also changed bone dramatically.

Below we have summarized what is known about parathyroid gland in evolution of vertebrates and aquatic mammals.

Parathyroid glands are indeed not observed in bony fish (Osteichthyes). Osteichthyes poses calcium-sensing PTH-secreting cells at their gills, but tetrapods have them evolved into parathyroid glands. However, no changes were reported between ultrastructural organization of parathyroid glands between amphibians and fully terrestrial tetrapods, such as reptiles, avian species and mammals (reviewed by Isono H et al., Histol Histopath 1990, 5:95-112). In relation to cetaceans, their transition back to water definitely associated not only with structural adaptation of the skeleton, but also global change in the bone-resistance to mechanical unloading. At the same time parathyroid glands in cetaceans are largely similar to those in terrestrial mammals, with exception that oxyphil cells have not been reported in cetaceans to our knowledge (Kwiecinski GG et al., The American Journal of Anatomy, 1987, 178:421427; Tsuchiya T et al., Tohoku Journal of Agricultural Research, 1983, 33:146-151; Kamiya T et al., Sci Rep Whales Res Inst 1978, 30:281-284; Hayakawa D et al., General and Comparative Endocrinology, 1998, 110:58-66). These cells have unknown function, but theoretically might be a source of PTHrP, as PTHrP is detected in oxyphil cells at high amount (Ritter CS, et al., The Journal of Clinical Endocrinology and Metabolism, 2012, 97:E1499-E1505). At the same time, it has to be admitted that various terrestrial species do not have oxyphil cells, such as rats or chicken (Altenähr E. Curr Top Pathol 1972; 56: 1-54).

Thus, parathyroid hormone is not likely participating in SOC development and we decided not to include this information into the Discussion. However, there are data suggesting that thyroid hormone is involved in SOC development and we have incorporated those into the Discussion (ninth paragraph).

We have also softened our conclusions discussing that other factors and systems can also contribute to evolution of the SOC (Discussion, seventh paragraph).

3) The gait pattern of jerboa that showed in the video clip doesn't support that the jerboa only uses its front limbs for crawling during the first 2 weeks of age. Instead, they seem to use their hind limbs for crawling more.

We apologize for doubtful videos presented. Due to ongoing pandemic and lockdown of UCSD (where jerboa colony is housed) we are not able to collect more videos and quantify the use of different limbs during jerboa early postnatal life. However, we have got in contact with Prof. David Eilam (Tel-Aviv University), who studied jerboa intensively in 90^th^. He also does not have ongoing colony of these animals but supplied us with the video showing that jerboas are using only forelimbs in their early life (now added as Figure 3—video 2). We have also referred to his previous work where such a gait acquisition pattern was documented in more details (Eilam and Shefer, 1997, the text is slightly adjusted accordingly, subsection “Comparative analysis of animals with specialized extremities”). We hope this new information would convince the reviewers that jerboas are utilizing their forelimbs much early than their hindlimbs.

4) There is a concern regarding the comparison between 10-day old rats and 30-day-old mice. The rodents are under fast growing during the first postnatal month. Although the indentation of cartilage of these two animal models is similar, the development of skeletal tissue such as muscle, ligaments may be different. Importantly, the intrinsic biological and/or mechanical properties, such as the proliferation rate or matrix protein synthesize capabilities of the chondrocytes in rat at P10 is likely distinctive to mouse at P30. In the Figure 5, it clearly shows that the cell number of the growth plate in the rat model in much more than the mouse model. Whether this characteristic affect the outcome?

In this test system we compare mouse and rat bones stripped off the muscles and ligaments. For intrinsic mechanical properties we compared the entire cartilage, which includes extracellular matrix, and found no differences by nanoindentation (cantilever probe is 45 micrometers in diameter). Proliferation rate is likely different but since the mechanical stress primarily affects hypertrophic chondrocytes, but not proliferative chondrocytes (Figure 6A-I and Figure 6—figure supplement 1E-F), this should not influence the conclusions drawn.

The number of cells in rat and mouse growth plate is indeed different. To take this into account the data are presented as the percentage of cells undergoing apoptosis.

We further supported this test model by a variety of in vivo experiments (Figure 8 and Figure 9B) and mathematical modeling, which predictions are in line with the observations made on this test model (i.e., stiffness of the epiphysis in Figure 5—figure supplement 1C-J and mathematical predictions of deflection Figure 1F).

To address the reviewers’ concern experimentally, we took a different approach and applied the same pressure (load normalized on cross-section area) on rat tibia at different developmental stages, i.e., postnatal day 5 (P5), P8, P12, P15 and P22, and analyzed correlation between load-induced cell death of hypertrophic chondrocytes and SOC development. This experiment revealed a clear protection of chondrocytes executed by the SOC (new Figure 7).

Thus, this new model leads to the same outcome as the early (mouse/rat SOC±) ex vivo model and together with in vivo models and mathematical predictions all pointing toward the same conclusion may provide an additional cross-validation. We have also clarified in the manuscript that SOC± models cannot be considered solely without potential cofounding variables (subsection “Functional experiments with two model systems: the SOC protects epiphyseal chondrocytes”).

We hope that this new experiment and considerations above address the reviewers concern.

5) The FEA simulation model is relatively simplified. For example, they assume both of the surfaces were oval and materials in all subdomains were considered to be linearly elastic, which brings concerns on the FEA to reflect the real mechanical situation.

The shapes in our FEA simulation model are simplified purposely to reveal the greatest conceptual differences between SOC+ and SOC- structures. We based our modelling on the previous key works in this field, i.e., focused on epiphyseal structures (Carter and Wong, 1988 and Carter, MikiĆ, and Padian, 1998).

The variety of SOC shapes found in nature is virtually endless and if there are specific forms, which reviewers are interested in, we are happy to model those forms as well. Otherwise we prefer to keep the most simplified forms to highlight the conceptual differences made by the SOC in its simplest form, which in fact is common for humans’ and rodents’ bones.

In relation to linear elasticity, it is originally developed by Hayes and co-authors (J of Biomechanics 1972, 5:541-551) and since then widely used in numerous publications (i.e., 956 citations) including aforementioned works by Carter. So, we employed this model to be able to align our data with the previous research on the epiphysis biomechanics (Carter and Wong, 1988 and Carter, MikiĆ and Padian, 1998). However, the reviewers are right, and over the recent years a biphasic model for cartilage (developed by Mow et al., J of Biomechanical Engineering, 1980, 102:73-84) has become more widely accepted. This biphasic model is particularly used for analysis of articular cartilage and various cartilage implants developed in connection to tissue engineering efforts for repair of articular cartilage.

Thus, we originally selected the linear elasticity model, reasoning that aligning our data with previous works focused on the SOC is more important than following more precise modeling focused on engineered tissue implants. We hope the reviewers will kindly agree with these considerations.

6) The influence of genetic modulation or axitinib treatment on changes of phenotype cannot be excluded if no evidence shows that the biological properties of the chondrocytes do not change in the axitinib or Prx-Cre: Gsa model.

We are slightly puzzled what biological properties are in question. We have extensively characterized the effect of axitinib on chondrocytes recently (Newton et al., 2019, Supplementary Table 1) and found that it does not affect chondrocyte proliferation.

In the context of this manuscript we have assumed that the effect on mechanical properties might be in question here. To address this, we have treated mice with axitinib exactly as before and have measured mechanical properties of cartilage by nanoindentation (cantilever head diameter 45 micrometers). We did not find significant differences in the stiffness of cartilage comparing vehicle and axitinib treated mice. Data are presented in text (subsection “Functional experiments with physiological systems: the SOC protects epiphyseal chondrocytes”).

Unfortunately we cannot do similar experiment with Prx1-Cre:Gsa mice since the strain is in Boston and the nanoindenter is in Moscow, and it is currently non-feasible to transfer the strain to Moscow.

We hope that the new data presented as well as previously-published analysis (Newton et al., 2019, Supplementary Table 1) satisfy the reviewers’ concern.